# Minimax-Optimal Multi-Agent RL in Markov Games With a Generative Model

**Gen Li**
UPenn

**Yuejie Chi**
CMU

**Yuting Wei**
UPenn

**Yuxin Chen**
UPenn

## Abstract

This paper studies multi-agent reinforcement learning in Markov games, with the goal of learning Nash equilibria or coarse correlated equilibria (CCE) sample-optimally. All prior results suffer from at least one of the two obstacles: the curse of multiple agents and the barrier of long horizon, regardless of the sampling protocol in use. We take a step towards settling this problem, assuming access to a flexible sampling mechanism: the generative model. Focusing on non-stationary finite-horizon Markov games, we develop a fast learning algorithm called Q-FTRL and an adaptive sampling scheme that leverage the optimism principle in online adversarial learning (particularly the Follow-the-Regularized-Leader (FTRL) method). Our algorithm learns an $\varepsilon$-approximate CCE in a general-sum Markov game using

$$\widetilde{O}\left(\frac{H^4 S \sum_{i=1}^m A_i}{\varepsilon^2}\right)$$

samples, where $m$ is the number of players, $S$ indicates the number of states, $H$ is the horizon, and $A_i$ denotes the number of actions for the $i$-th player. This is minimax-optimal (up to log factor) when $m$ is fixed. When applied to two-player zero-sum Markov games, our algorithm provably finds an $\varepsilon$-approximate Nash equilibrium with a minimal number of samples. Along the way, we derive a refined regret bound for FTRL that makes explicit the role of variance-type quantities, which might be of independent interest.

## 1 Introduction

The thriving field of multi-agent reinforcement learning (MARL) studies how a group of interacting agents make decisions autonomously in a shared dynamic environment [80]. The recent developments in game playing [66, 9], self-driving vehicles [58], and multi-robot control [45] are prime examples of MARL in action. In practice, there is no shortage of situations where the agents involved have conflict of interest, and they have to act competitively in order to promote their own benefits (possibly at the expense of one another). Scenarios of this kind are frequently modeled via Markov games (MGs) [59, 42], a framework that has been a fruitful playground to formalize and stimulate the studies of competitive MARL.

In view of the irreconcilable competition between individual players, solutions of competitive MARL normally take the form of certain equilibrium strategy profiles, which are perhaps best epitomized by the concept of Nash equilibrium (NE) [49]. In a Nash equilibrium, no gain can be realized through a unilateral change — assuming no coordination between players — and hence no player has incentives to deviate from her current strategy/policy. A myriad of research has been conducted surrounding NE, which spans various aspects like existence, learnability, computational hardness, and algorithm design, among others [59, 20, 12, 53, 52, 22, 42, 28, 50, 33]. Given that finding NE is notoriously expensive in general (except for special cases like two-player zero-sum MGs) [20, 21], several more tractable solution concepts have emerged in the studies of game theory and MARL, a

36th Conference on Neural Information Processing Systems (NeurIPS 2022).

prominent example being the coarse correlated equilibrium (CCE) [47]. A key compromise made in the CCE is that it permits the players to act in an coordinated fashion, which contrasts sharply with the absence of coordination in the definition of NE.

One critical challenge impacting modern MARL applications is data efficiency. The players involved often have minimal knowledge about how the environment responds to their actions, and have to learn the dynamics and preferable actions by probing the unknown environment. For MARL to expand into applications with enormous dimensionality and long planning horizon, the learning algorithms must manage to make efficient use of the collected data. Nevertheless, how to learn NE and/or CCE with optimal sample complexity remains by and large unsettled even when it comes to the most basic setting: two-player zero-sum Markov games, as we shall discuss below.

**Example: inadequacy in learning two-player zero-sum Markov games.** To facilitate concrete comparisons, let us review two representative algorithms aimed at learning NE in two-player zero-sum MGs. These algorithms have been studied under two drastically different sampling protocols, and we shall discuss the shortfalls of the cutting-edge sample complexity results. In a two-player zero-sum MG, we denote by $S$ the number of states and $H$ the horizon or effective horizon, whereas $A_1$ and $A_2$ denote respectively the number of actions for the max-player and the min-player.

- *Model-based methods under either a generative model or online exploration.* Assuming access to a generative model (so that one can sample arbitrary state-action tuples), Zhang et al. [79] investigated a natural model-based algorithm, which performs planning (e.g., value iteration) on an empirical MG derived from samples produced non-adaptively by the generative model. Focusing on *stationary* discounted infinite-horizon MGs, their algorithm finds an $\varepsilon$-approximate NE with

$$\widetilde{O}\left(\frac{H^3 S A_1 A_2}{\varepsilon^2}\right) \text{ samples.} \tag{1}$$

In parallel, Liu et al. [43] studied *non-stationary* finite-horizon MGs with online exploration, and obtained similar sample complexity bounds, i.e.,

$$\widetilde{O}\left(\frac{H^4 S A_1 A_2}{\varepsilon^2}\right) \text{ samples} \qquad \text{or} \qquad \widetilde{O}\left(\frac{H^3 S A_1 A_2}{\varepsilon^2}\right) \text{ episodes} \tag{2}$$

for learning an $\varepsilon$-approximate NE. While these bounds achieve minimax-optimal dependency on the horizon $H$, a major drawback emerges — commonly referred to as the curse of multiple agents; namely, these results scale proportionally with the total number of *joint actions* (i.e., $\prod_{1 \leq i \leq 2} A_i$), a quantity that blows up exponentially with the number of players.

- *V-learning for online exploration settings.* Focusing on online exploration settings, Bai et al. [5], Jin et al. [31] proposed an algorithm called V-learning that leverages the advances in online adversarial learning (e.g., adversarial bandits) to circumvent the curse of multiple agents. This algorithm provably yields an $\varepsilon$-approximate NE in non-stationary finite-horizon MGs using

$$\widetilde{O}\left(\frac{H^6 S(A_1 + A_2)}{\varepsilon^2}\right) \text{ samples} \qquad \text{or} \qquad \widetilde{O}\left(\frac{H^5 S(A_1 + A_2)}{\varepsilon^2}\right) \text{ episodes,} \tag{3}$$

which effectively brings down the sample size scaling (2) from $A_1 A_2$ (i.e., the number of joint actions) to $A_1 + A_2$ (i.e., the sum of individual actions). It is worth pointing out, however, that this theory appears sub-optimal in terms of the horizon dependency, as it is a factor of $H^2$ above the minimax lower bound.

**Key issues and our main contributions.** While the above summary focuses on two-player zero-sum MGs, it unveils a fundamental issue surrounding the sample efficiency of learning equilibria; that is, all existing results in this front — irrespective of the sampling mechanism in use — fall short of overcoming at least one of the two major hurdles: (i) the *curse of multiple agents*, and (ii) the *barrier of long horizon*. A natural question to pose is:

> **Question:** *can we learn a Nash equilibrium in a two-player zero-sum Markov game in a sample-optimal and computation-efficient fashion?*

To settle this favorably, both of the above hurdles need to be crossed simultaneously. Moving beyond two-player zero-sum MGs, it is not surprising to see that general-sum multi-player MGs have to grapple with the aforementioned two hurdles as well. Thus, the following question also comes into mind when learning CCE (a compromise due to the general intractability of learning NE):

**Question:** *can we learn a coarse correlated equilibrium in a multi-player general-sum Markov game in a sample-optimal and computation-efficient fashion?*

Note that these questions remain open regardless of the sampling scheme in use.

This paper takes a first step towards solving the problem by assuming access to the most flexible sampling protocol: the generative model (or simulator). In stark contrast to the single-agent case where uniform sampling of all state-action pairs suffices [3, 38], the multi-agent scenario requires one to take samples intelligently and adaptively, a crucial step to avoid inefficient use of data (otherwise one cannot hope to break the curse of multiple agents). With the aim of computing an $\varepsilon$-approximate equilibrium in a *non-stationary* finite-horizon MG, we come up with a computationally efficient learning algorithm (accompanied by an adaptive sampling strategy) that accomplishes this goal with no more than

$$
\begin{cases}
\widetilde{O}\left(\frac{H^4 S (A_1 + A_2)}{\varepsilon^2}\right) \text{ samples} & \text{(learning } \varepsilon\text{-NE in two-player zero-sum MGs)} \\
\widetilde{O}\left(\frac{H^4 S \left(\sum_{i=1}^m A_i\right)}{\varepsilon^2}\right) \text{ samples} & \text{(learning } \varepsilon\text{-CCE in multi-player general-sum MGs)}
\end{cases}
\tag{4}
$$

drawn from the generative model. Encouragingly, this sample complexity bound matches the minimax lower limit (up to a logarithmic factor) as long as the number of players $m \geq 2$ is a fixed constant or grows only logarithmically in problem parameters. Our sample complexity theory is valid for the full $\varepsilon$-range (i.e., any $\varepsilon \in (0, H]$); this unveils that no burn-in cost is needed for our algorithm to achieve sample optimality, which lends itself well to sample-hungry applications.

The proposed algorithm is inspired by two key algorithmic ideas in RL and bandit literature: (i) optimism in the face of uncertainty (by leveraging upper confidence bounds (UCBs) in value estimation), and (ii) online and adversarial learning (particularly the Follow-the-Regularized-Leader (FTRL) algorithm). Note that the optimal design of bonus terms — typically based on certain data-driven variance estimates — is substantially more challenging than the single-agent case, as it requires intricate adaptation in response to the policy changes of one another as well as compatibility with the FTRL dynamics. Two points are worth emphasizing (which will be made precise later on):

- The efficacy of FTRL in breaking the curse of multiple agents has been illustrated in Jin et al. [31], Song et al. [63], Mao and Başar [44]. To improve horizon dependency, one needs to exploit connections between the performance of FTRL and certain variances. Towards this, we develop a refined regret bound for FTRL that unveils the role of variance-style quantities, which was previously unavailable.

- The bonus terms entail Bernstein-style variance estimates that mimic the variance-style quantities appearing in our refined FTRL regret bounds, and are carefully chosen so as to ensure certain decomposability over steps. This is crucial in optimizing the horizon dependency.

Additionally, the policy returned by our algorithm is Markovian (i.e., the action selection probability depends only on the current state $s$ and step $h$), and the algorithm can be carried out in a decentralized manner without the need of directly observing the opponents' actions.

**Notation.** Let us also gather several convenient notation that shall be used multiple times. For any positive integer $n$, we write $[n] := \{1, \cdots, n\}$. We shall abuse notation and let $1$ and $0$ denote the all-one vector and the all-zero vector, respectively. For a sequence $\{\alpha_k\}_{k \geq 1} \subseteq (0, 1]$, we define

$$
\alpha_i^k := \begin{cases}
\alpha_i \prod_{j=i+1}^k (1 - \alpha_j), & \text{if } 0 < i < k \\
\alpha_k, & \text{if } i = k
\end{cases}
\tag{5}
$$

for any $1 \leq i \leq k$. For a given vector $x \in \mathbb{R}^{SA}$ (resp. $y \in \mathbb{R}^{SAB}$), we denote by $x(s, a)$ (resp. $y(s, a, b)$) the entry of $x$ (resp. $y$) associated with the state-action combination $(s, a)$ (resp. $(s, a, b)$), as long as it is clear from the context. Next, consider any two vectors $a = [a_i]_{1 \leq i \leq n}$ and $b = [b_i]_{1 \leq i \leq n}$. We use $a \leq b$ (resp. $a \geq b$) to indicate that $a_i \geq b_i$ (resp. $a_i \leq b_i$) holds for all $i$; we allow scalar functions to take vector-valued arguments in order to denote entrywise operations (e.g., $a^2 = [a_i^2]_{1 \leq i \leq n}$ and $a^4 = [a_i^4]_{1 \leq i \leq n}$); and we denote by $a \circ b = [a_i b_i]_{1 \leq i \leq n}$ the Hadamard product. For a finite set $\mathcal{A} = \{1, \cdots, A\}$, we denote by $\Delta(\mathcal{A}) = \{x \in \mathbb{R}^A \mid \sum_i x_i = 1; x \geq 0\}$ the probability simplex over $\mathcal{A}$. For any function $f$ with domain $\mathcal{A}$ (or $\mathcal{B}$), we adopt the notation

$$
\mathbb{E}_\pi[f] := \sum_a \pi(a) f(a) \qquad \text{and} \qquad \mathsf{Var}_\pi(f) := \sum_a \pi(a) \big(f(a) - \mathbb{E}_\pi[f]\big)^2. \tag{6}
$$

## 2 Background and models

In this section, we introduce the basics for Markov games, as well as the solution concepts of Nash equilibrium and coarse correlated equilibrium.

**Markov games.** A non-stationary finite-horizon *multi-player general-sum Markov game*, denoted by $\mathcal{MG} = \{\mathcal{S}, \{\mathcal{A}_i\}_{1 \le i \le m}, H, P, r\}$, involves $m$ players competing against each other, and consists of several key elements to be formalized below. Recall that $\Delta(\mathcal{S})$ represents the probability simplex over the set $\mathcal{S}$.

- $\mathcal{S} = \{1, \cdots, S\}$ is the state space of the shared environment, which comprises $S$ different states.
- For each $1 \le i \le m$, let $\mathcal{A}_i = \{1, \cdots, A_i\}$ represent the action space of the $i$-th player, which contains $A_i$ different actions. Here and below, we denote

$$\mathcal{A} := \mathcal{A}_1 \times \cdots \times \mathcal{A}_m \qquad \text{and} \qquad \mathcal{A}_{-i} := \prod_{j:j \ne i} \mathcal{A}_j \quad (1 \le i \le m). \tag{7}$$

  Throughout the paper, we shall often use the boldface letter $\boldsymbol{a} \in \mathcal{A}$ (resp. $\boldsymbol{a}_{-i} \in \mathcal{A}_{-i}$) to denote a joint action profile of all players (resp. a joint action profile excluding the $i$-th player's action).

- $H$ stands for the horizon length of the Markov game.
- $P = \{P_h\}_{1 \le h \le H}$ — with $P_h : \mathcal{S} \times \mathcal{A} \to \Delta(\mathcal{S})$ — denotes the probability transition kernel of $\mathcal{MG}$. Namely, for any $(s, \boldsymbol{a}, h, s') \in \mathcal{S} \times \mathcal{A} \times [H] \times \mathcal{S}$, we let $P_h(s' \,|\, s, \boldsymbol{a})$ indicate the probability of $\mathcal{MG}$ transitioning from state $s$ to state $s'$ at step $h$ when the joint action profile taken by the players is $\boldsymbol{a}$.
- $r = \{r_{i,h}\}_{1 \le h \le H, 1 \le i \le m}$ — with $r_{i,h} : \mathcal{S} \times \mathcal{A} \to [0,1]$ — represents the (deterministic) reward function. Namely, for any $(s, \boldsymbol{a}, h) \in \mathcal{S} \times \mathcal{A} \times [H]$, $r_{i,h}(s, \boldsymbol{a})$ stands for the immediate reward the $i$-th player gains in state $s$ at step $h$, if the joint action profile is $\boldsymbol{a}$. Here and throughout, we assume normalized rewards in the sense that $r_{i,h}(s, \boldsymbol{a}) \in [0,1]$ for any $(s, \boldsymbol{a}, h, i) \in \mathcal{S} \times \mathcal{A} \times [H] \times [m]$.

As an important special case, a *two-player zero-sum Markov game* — denoted by $\mathcal{MG} = \{\mathcal{S}, \{\mathcal{A}_1, \mathcal{A}_2\}, H, P, r\}$ — satisfies $r_{2,h} = -r_{1,h}$ for all $h \in [H]$. Following the convention, we assume that $r_{1,h} \ge 0$ for all $h \in [H]$,[1] and refer to the first (resp. second) player as the max-player (resp. the min-player).

**Markov policies.** This paper focuses on the class of Markov policies, such that the action selection strategies of the players are determined by the current state $s$ and the step number $h$, without depending on previously visited states. To begin with, let $\pi_i = \{\pi_{i,h}\}_{1 \le h \le H}$ represent the policy of the $i$-th player. Here, $\pi_{i,h}(\cdot \,|\, s) \in \Delta(\mathcal{A}_i)$ for any $(s, h) \in \mathcal{S} \times [H]$, where $\pi_{i,h}(a \,|\, s)$ indicates the probability of the $i$-th player selecting action $a$ in state $s$ at step $h$. The joint Markov policy can be defined analogously: we let $\pi = (\pi_1, \ldots, \pi_m) : \mathcal{S} \times [H] \to \Delta(\mathcal{A})$ represent a joint Markov policy of all players, where the joint actions of all players in state $s$ and step $h$ are chosen according to the distribution specified by $\pi_h(\cdot \,|\, s) = (\pi_{1,h}, \ldots, \pi_{m,h})(\cdot \,|\, s) \in \Delta(\mathcal{A})$. For any given joint policy $\pi$, we employ $\pi_{-i}$ to represent the policies of all but the $i$-th player, and let $\pi_{-i,h}$ denote the policies of all but the $i$-th player at step $h$. All policies are assumed throughout to be Markovian, except our brief remarks on non-Markovian policies in Section 3.2.

Additionally, a joint policy $\pi$ is said to be a *product policy* if $\pi_1, \ldots, \pi_m$ are executed in a statistically independent manner (namely, under policy $\pi$ the players take actions independently), and we shall adopt the notation $\pi = \pi_1 \times \cdots \times \pi_m$ to indicate that $\pi$ is a product policy.

**Value functions.** Consider a Markovian trajectory $\{(s_h, \boldsymbol{a}_h)\}_{1 \le h \le H}$, where $s_h \in \mathcal{S}$ is the state at step $h$ and $\boldsymbol{a}_h \in \mathcal{A}$ is the joint action profile at step $h$. For any given joint policy $\pi$ and any step $h \in [H]$, we define the value function $V_{i,h}^\pi : \mathcal{S} \to \mathbb{R}$ of the $i$-th player under policy $\pi$ as follows:

$$V_{i,h}^\pi(s) := \mathbb{E}\left[\sum_{t=h}^{H} r_{i,t}(s_t, \boldsymbol{a}_t) \,\mid\, s_h = s\right], \qquad \forall s \in \mathcal{S}, \tag{8}$$

---

[1]The careful reader might immediately note that $r_{2,h} \le 0$, thus falling outside our assumed range for the reward function. This, however, can be easily addressed by enforcing a positive global shift to $r_{2,h}$ without changing the learning process.

where the expectation is taken over the Markovian trajectory $\{(s_h, \boldsymbol{a}_h)\}$ with the $m$ players jointly executing policy $\pi$; that is, conditional on $s_h$, we draw $\boldsymbol{a}_h \sim \pi_h(\cdot \,|\, s_h)$ and then $s_{h+1} \sim P_h(\cdot \,|\, s_h, \boldsymbol{a}_h)$.

In addition, consider the case where (i) all but the $i$-th player executes the joint policy $\pi_{-i}$ and (ii) the $i$-th player executes policy $\pi_i'$ *independently* from the other players; we shall denote by $V_{i,h}^{\pi_i' \times \pi_{-i}}$ the resulting value function under this joint policy $\pi_i' \times \pi_{-i}$. By optimizing over all $\pi_i'$, we can further define

$$V_{i,h}^{\star,\pi_{-i}}(s) := \max_{\pi_i' : \mathcal{S} \times [H] \to \Delta(\mathcal{A}_i)} V_{i,h}^{\pi_i' \times \pi_{-i}}(s), \qquad \forall (s, h, i) \in \mathcal{S} \times [H] \times [m]. \tag{9}$$

It is known that there exists at least one policy, denoted by $\pi_i^\star(\pi_{-i}) : \mathcal{S} \times [H] \to \Delta(\mathcal{A}_i)$ and commonly referred to as the *best-response policy*, that can simultaneously attain $V_{i,h}^{\star,\pi_{-i}}(s)$ for all $h \in [H]$ and all $s \in \mathcal{S}$. It is worth emphasizing that the best-response policy $\pi_i^\star(\pi_{-i})$ is the best among all policies of the $i$-th player executed independently of $\pi_{-i}$. Furthermore, if we freeze $\pi_{-i}$, then the Bellman optimality condition for the $i$-th player can be expressed as [8]

$$V_{i,h}^{\star,\pi_{-i}}(s) = \max_{a_i \in \mathcal{A}_i} \left\{ \mathop{\mathbb{E}}_{\boldsymbol{a}_{-i} \sim \pi_{-i,h}(\cdot|s)} \left[ r_{i,h}(s, \boldsymbol{a}) + \left\langle P_h(\cdot \,|\, s, \boldsymbol{a}), V_{i,h+1}^{\star,\pi_{-i}} \right\rangle \right] \right\} \tag{10}$$

for all $(s, h, i) \in \mathcal{S} \times [H] \times [m]$, where the joint action profile $\boldsymbol{a}$ is composed of $a_i$ for the $i$-th player and $\boldsymbol{a}_{-i}$ for the remaining ones.

**Equilibria of Markov games.** In a multi-agent Markov game, each player wishes to maximize its own value function. Due to the competing objectives, finding some sorts of equilibria — e.g., the Nash equilibrium [48] and the coarse correlated equilibrium [47, 2] — becomes a central topic in the studies of Markov games. Let us introduce these solution concepts below.

- *Nash equilibrium.* A product policy $\pi = \pi_1 \times \cdots \times \pi_m$ is said to be a *(mixed-strategy) Nash equilibrium* of $\mathcal{MG}$ if the following holds:

$$V_{i,1}^{\pi}(s) = V_{i,1}^{\star,\pi_{-i}}(s), \qquad \text{for all } (s, i) \in \mathcal{S} \times [m]. \tag{11}$$

  In other words, conditional on the opponents' current policy and the assumption that all players take actions *independently*, no player can harvest any gain by unilaterally deviating from its current policy.

- *Coarse correlated equilibrium.* A joint policy $\pi$ is said to be a coarse correlated equilibrium of $\mathcal{MG}$ if

$$V_{i,1}^{\pi}(s) \geq V_{i,1}^{\star,\pi_{-i}}(s), \qquad \text{for all } (s, i) \in \mathcal{S} \times [m]. \tag{12}$$

  While a CCE also ensures that no unilateral deviation (performed independently from others) is beneficial, its key distinction from the definition of NE lies in the fact that it allows the policy to be correlated across the players. Any NE of $\mathcal{MG}$ is, self-evidently, also a CCE.

In practice, it might be challenging to compute an "exact" equilibrium, and instead one would seek to find approximate solutions. Towards this end, we find it helpful to define the sub-optimality gap of a policy $\pi$ as follows (measured in an $\ell_\infty$-based manner)

$$\mathsf{gap}(\pi) := \max_{s \in \mathcal{S}} \mathsf{gap}(\pi; s), \tag{13a}$$

where

$$\mathsf{gap}(\pi; s) := \max_{1 \leq i \leq m} \left\{ V_{i,1}^{\star,\pi_{-i}}(s) - V_{i,1}^{\pi}(s) \right\}. \tag{13b}$$

With this sub-optimality measure in place, a *product* policy $\pi = \pi_1 \times \cdots \times \pi_m$ is said to be an $\varepsilon$-approximate NE — or more concisely, $\varepsilon$-Nash — if the resultant sub-optimality gap obeys $\mathsf{gap}(\pi) \leq \varepsilon$. Similarly, a joint (and possibly correlated) policy $\pi$ is said to be an $\varepsilon$-approximate CCE — or more concisely, $\varepsilon$-CCE — if $\mathsf{gap}(\pi) \leq \varepsilon$.

**Generative model / simulator.** In reality, we oftentimes do not have access to perfect descriptions (e.g., accurate knowledge of the transition kernel $P$) of the Markov game under consideration; instead, one has to learn the true model on the basis of data samples. When it comes to the data generating mechanism, this paper assumes access to a generative model (also called a simulator) [35, 34]:

in each call to the generative model, the learner can choose an arbitrary $(s, \boldsymbol{a}, h) \in \mathcal{S} \times \mathcal{A} \times [H]$ and obtain an independent sample generated based on the true transition kernel:

$$s' \sim P_h(\cdot \,|\, s, \boldsymbol{a}).$$

In words, a generative model facilitates query of arbitrary state-action-step tuples, which helps alleviate the sampling constraints arising in online episodic settings for exploration. The goal of the current paper is to compute an $\varepsilon$-approximate equilibrium (either NE or CCE) of $\mathcal{MG}$ with as few samples as possible, i.e., using a minimal number of calls to the generative model.

## 3 Sample-efficient learning with a generative model

In this section, we put forward an efficient algorithm aimed at learning an $\varepsilon$-approximate equilibrium with the assistance of a generative model, and demonstrate its sample optimality for the full $\varepsilon$-range.

### 3.1 Algorithm description

We now describe the proposed algorithm, which is inspired by the optimism principle and the FTRL algorithm for online/adversarial learning. Following the dynamic programming approach [8], our algorithm employs backward recursion from step $h = H$ back to $h = 1$; in fact, we shall finish the sampling and learning processes for step $h$ before moving backward to step $h - 1$. For each $h$, the $i$-th player calls the generative model for $K$ rounds, with each round drawing $SA_i$ independent samples; as a result, the total sample size is given by $KSH \sum_{i=1}^m A_i$. In what follows, let us first introduce some convenient notation that facilitates our exposition of the algorithm.

**Notation.** Consider any step $h \in [H]$, any player $i \in [m]$, and any data collection round $k \in [K]$. The algorithm maintains the following iterates, whose notation is gathered here with their formal definitions introduced later.

- $\widehat{V}_{i,h} \in \mathbb{R}^S$ represents the final estimate of the value function at step $h$ by the $i$-th player; in particular, we set $\widehat{V}_{i,H+1} = 0$.

- $Q_{i,h}^k \in \mathbb{R}^{SA_i}$ represents the Q-function estimate of the $i$-th player at step $h$ after the $k$-th round of data collection.

- $q_{i,h}^k \in \mathbb{R}^{SA_i}$ stands for a certain "one-step-look-ahead" Q-function estimate of the $i$-th player at step $h$ using samples collected in the $k$-th round.

- $r_{i,h}^k \in \mathbb{R}^{SA_i}$ denotes the sample reward vector for step $h$ received by the $i$-th player in the $k$-th round.

- $P_{i,h}^k \in \mathbb{R}^{SA_i \times S}$ denotes the empirical probability transition matrix for step $h$ constructed using the samples collected by the $i$-th player in the $k$-th round.

- $\beta_{i,h} \in \mathbb{R}^S$ denotes the bonus vector chosen by the $i$-th player at step $h$ during final value estimation.

- $\pi_{i,h}^k : \mathcal{S} \to \Delta(\mathcal{A}_i)$ denotes the policy iterate of the $i$-th player at step $h$ before the beginning of the $k$-th round of data collection; in particular, we set $\pi_{i,h}^1$ to be uniform, namely, $\pi_{i,h}^1(a_i \,|\, s) = 1/A_i$ for any $(s, a_i) \in \mathcal{S} \times \mathcal{A}_i$.

Crucially, the above objects are all constructed from the perspective of a single player, and hence resemble those needed to operate a "single-agent" MDP (as opposed to MARL). As such, the complexity of storing/updating the above objects only scales with the aggregate size of the individual action space, rather than the size of the product action space.

**Main steps of the proposed algorithm.** As mentioned above, our algorithm collects multiple rounds of independent samples for each $h$. In what follows, let us describe the proposed procedure for the $i$-th player in the $k$-th round for step $h$.

1. *Sampling and model estimation.* For each $(s, a_i) \in \mathcal{S} \times \mathcal{A}_i$, draw an *independent* sample as follows

$$s'_{k,h,s,a_i} \sim P_h\big(\,\cdot\,|\, s, \boldsymbol{a}(k,h,s,a_i)\big) \qquad \text{and} \qquad r_{k,i,h,s,a_i} = r_{i,h}\big(s, \boldsymbol{a}(k,h,s,a_i)\big), \qquad \text{(14a)}$$

where $\boldsymbol{a}(k, h, s, a_i) = [a_j(k, h, s, a_i)]_{1 \leq j \leq m} \in \mathcal{A}$ consists of independent individual actions drawn from

$$a_j(k, h, s, a_i) \overset{\text{ind.}}{\sim} \pi_{j,h}^k(\cdot \,|\, s) \quad (j \neq i) \qquad \text{and} \qquad a_i(k, h, s, a_i) = a_i. \tag{14b}$$

These samples are then employed to construct the sample reward vector $r_{i,h}^k \in \mathbb{R}^{SA_i}$ and empirical probability transition kernel $P_{i,h}^k \in \mathbb{R}^{SA_i \times S}$ such that

$$r_{i,h}^k(s, a_i) = r_{k,i,h,s,a_i} \qquad \text{and} \qquad P_{i,h}^k(s' \,|\, s, a_i) = \begin{cases} 1, & \text{if } s' = s'_{k,h,s,a_i} \\ 0, & \text{else} \end{cases} \tag{14c}$$

for all $(s, a_i, s') \in \mathcal{S} \times \mathcal{A}_i \times \mathcal{S}$. Note that the $i$-th player only needs to compute (14c), without the need of directly observing the other players' actions.

2. *Q-function estimation.* Following the dynamic programming approach, we first compute the "one-step-look-ahead" Q-function estimate as follows

$$q_{i,h}^k = r_{i,h}^k + P_{i,h}^k \widehat{V}_{i,h+1}. \tag{15}$$

We then adopt the update rule of Q-learning:

$$Q_{i,h}^k = (1 - \alpha_k) Q_{i,h}^{k-1} + \alpha_k q_{i,h}^k, \tag{16}$$

where $0 < \alpha_k < 1$ is the learning rate. Applying (16) recursively and using the quantities defined in (5), we easily arrive at the following expansion:

$$Q_{i,h}^k = \sum_{j=1}^k \alpha_j^k q_{i,h}^j. \tag{17}$$

3. *Policy updates.* Once the Q-estimates are updated, we adopt the exponential weights strategy to update the policy iterate of the $i$-th player as follows

$$\pi_{i,h}^{k+1}(a_i \,|\, s) = \frac{\exp\left(\eta_{k+1} Q_{i,h}^k(s, a_i)\right)}{\sum_{a' \in \mathcal{A}_i} \exp\left(\eta_{k+1} Q_{i,h}^k(s, a')\right)}, \qquad \forall (s, a_i) \in \mathcal{S} \times \mathcal{A}_i, \tag{18}$$

where $\eta_{k+1} > 0$ is another learning rate associated with policy updates (to be specified shortly). In fact, this subroutine implements the Follow-the-Regularized-Leader strategy [56]:

$$\pi_{i,h}^{k+1}(\cdot \,|\, s) = \arg \min_{\mu \in \Delta(\mathcal{A}_i)} \left\{ -\left\langle \mu, Q_{i,h}^k(s, \cdot) \right\rangle + \frac{1}{\eta_{k+1}} F(\mu) \right\}, \tag{19}$$

where the regularizer $F(\cdot)$ is chosen to be the negative entropy function $F(\mu) := \sum_{a \in \mathcal{A}_i} \mu(a) \log\left(\mu(a)\right)$.

After carrying out $K$ rounds of the above procedure, our final policy estimate $\widehat{\pi} : \mathcal{S} \times [H] \to \Delta(\mathcal{A})$ and the value estimate $\widehat{V}_{i,h} : \mathcal{S} \to \mathbb{R}$ for step $h$ are taken respectively to be

$$\widehat{V}_{i,h}(s) = \min \left\{ \sum_{k=1}^K \alpha_k^K \left\langle \pi_{i,h}^k(\cdot \,|\, s), q_{i,h}^k(s, \cdot) \right\rangle + \beta_{i,h}(s), \ H - h + 1 \right\} \qquad \text{and} \tag{20a}$$

$$\widehat{\pi}_h(\boldsymbol{a} \,|\, s) = \sum_{k=1}^K \alpha_k^K \prod_{i=1}^m \pi_{i,h}^k(a_i \,|\, s) \tag{20b}$$

for any $(s, \boldsymbol{a} = [a_1, \ldots, a_m]) \in \mathcal{S} \times \mathcal{A}$, where $\{\alpha_k^K\}$ is defined in (5) and $\beta_{i,h}(s) \geq 0$ is some bonus term (taking the form of some data-driven upper confidence bound) to be specified momentarily. It is worth pointing out that the final policy (20b) takes the form of *a mixture of product policies*. In the special case of two-player zero-sum MGs, we can alternatively output a product policy

$$\text{(two-player zero-sum MGs)} \qquad \widehat{\pi} = \widehat{\pi}_1 \times \widehat{\pi}_2, \tag{21}$$

where for each $i = 1, 2$, we take $\widehat{\pi}_i = \{\widehat{\pi}_{i,h}\}_{1 \leq h \leq H}$ with $\widehat{\pi}_{i,h} = \sum_{k=1}^K \alpha_k^K \pi_{i,h}^k$.

---
**Algorithm 1:** Q-FTRL.
---
1 **Input:** number of rounds $K$ for each step, learning rates $\{\alpha_k\}$ (cf. (22)) and $\{\eta_{k+1}\}$ (cf. (23)).

    // set initial value estimates to 0, and initial policies to uniform
        distributions.

2 **Initialize:** for any $i \in [m]$ and any $(s, a_i, h) \in \mathcal{S} \times \mathcal{A}_i \times [H]$, set $\widehat{V}_{i,H+1}(s) = Q_{i,h}^0(s, a_i) = 0$
    and $\pi_{i,h}^1(a_i \,|\, s) = 1/A_i$.

3 **for** $h = H$ **to** 1 **do**

4    **for** $k = 1$ **to** $K$ **do**

5       **for** $i = 1$ **to** $m$ **do**

          // draw independent samples, and construct empirical models.

6          $(r_{i,h}^k, P_{i,h}^k) \leftarrow$ sampling$(i, h, \pi_h^k = \{\pi_{j,h}^k\}_{j\in[m]})$.   /\* see Algorithm 2.
         \*/

          // update Q-estimates with upper confidence bounds.

7          Compute $q_{i,h}^k = r_{i,h}^k + P_{i,h}^k \widehat{V}_{i,h+1}$, and update $Q_{i,h}^k$ according to (16).

          // update policy estimates using FTRL.

8          Update $\pi_{i,h}^{k+1}$ according to (18).

    // output the final value estimate for step $h$.

9    **for** $i = 1$ **to** $m$ **do**

10       Update $\widehat{V}_{i,h}$ according to (20a), where $\beta_{i,h}$ is given in (24).

11 **if** $\mathcal{MG}$ *is a two-player zero-sum Markov game* **then**

12    **output:** $\widehat{\pi}_1 \times \widehat{\pi}_2$, where for any $i = 1, 2$, $\widehat{\pi}_i = \{\widehat{\pi}_{i,h}\}_{1 \le h \le H}$ with $\widehat{\pi}_{i,h} = \sum_{k=1}^K \alpha_k^K \pi_{i,h}^k$.

13 **if** $\mathcal{MG}$ *is a multi-player general-sum Markov game* **then**

14    **output:** $\widehat{\pi} = \{\widehat{\pi}_h\}_{1 \le h \le H}$, where $\widehat{\pi}_h = \sum_{k=1}^K \alpha_k^K \left(\pi_{1,h}^k \times \cdots \times \pi_{m,h}^k\right)$.
---

The whole procedure is summarized in Algorithm 1.

**Choices of learning rates.** Thus far, we have not yet specified the two sequences of learning rates, which we describe now. The learning rates associated with Q-function updates are set to be rescaled linear, namely,

$$\alpha_k = \frac{c_\alpha \log K}{k - 1 + c_\alpha \log K}, \qquad k = 1, 2, \dots \tag{22}$$

for some constant $c_\alpha \ge 24$. In addition, the learning rates associated with policy updates are chosen to be:

$$\eta_{k+1} = \sqrt{\frac{\log K}{\alpha_k H}}, \qquad k = 1, 2, \dots \tag{23}$$

**Choices of bonus terms.** It remains to specify the bonus terms, which are selected based on fairly intricate upper confidence bounds. This constitutes a key — and perhaps the most challenging — component of our algorithm design. Specifically, we take

$$\beta_{i,h}(s) = c_{\mathsf{b}} \sqrt{\frac{\log^3\left(\frac{KS \sum_i A_i}{\delta}\right)}{KH} \sum_{k=1}^K \alpha_k^K \left\{ \mathsf{Var}_{\pi_{i,h}^k(\cdot|s)}\left(q_{i,h}^k(s, \cdot)\right) + H \right\}} \tag{24}$$

for any $(i, s, h) \in [m] \times \mathcal{S} \times [H]$, where $c_{\mathsf{b}} > 0$ is some sufficiently large constant; see also (6) for the definition of the variance-style quantity. As in previous works, the bonus terms, which are chosen carefully in a data-driven fashion, need to compensate for the uncertainty incurred during the estimation process.

### 3.2 Main results

As it turns out, the proposed algorithm is tractable and provably sample-efficient. We begin by characterizing its sample complexity when learning Nash equilibria in two-player zero-sum MGs,

---

**Algorithm 2:** Auxiliary function $\mathtt{sampling}\big(i, h, \pi_h = \{\pi_{j,h}\}_{j \in [m]}\big)$.

---

**1** **Initialize:** $\overline{r} = 0 \in \mathbb{R}^{SA_i}$, and $\overline{P} = 0 \in \mathbb{R}^{SA_i \times S}$.

**2** **for** $(s, a_i) \in \mathcal{S} \times \mathcal{A}_i$ **do**

**3** $\quad$ Draw an independent sample from the generative model: $s'_{s,a_i} \sim P_h\big( \cdot \mid s, \boldsymbol{a}(s, a_i)\big)$, where $\boldsymbol{a}(s, a_i) = [a_j(s, a_i)]_{1 \le j \le m}$ is composed of independent individual actions drawn from

$$a_j(s, a_i) \overset{\text{ind.}}{\sim} \pi_{j,h}(\cdot \mid s) \quad (j \ne i) \qquad \text{and} \qquad a_i(s, a_i) = a_i. \tag{25}$$

**4** $\quad$ Set $\overline{r}(s, a_i) = r_{i,h}\big(s, \boldsymbol{a}(s, a_i)\big)$ and $\overline{P}\big(s'_{s,a_i} \mid s, a_i\big) = 1$.

**5** **Return:** $(\overline{r}, \overline{P})$.

---

and then shift attention to learning CCE in multi-player general-sum MGs (given the intractability of learning NEs in general).

**Theorem 1** (NE for two-player zero-sum MGs)**.** *Consider a two-player zero-sum Markov game, and consider any $\varepsilon \in (0, H]$ and any $0 < \delta < 1$. Suppose that*

$$K \ge \frac{c_{\mathsf{k}} H^3 \log^4 \big( \frac{KS(A_1 + A_2)}{\delta} \big)}{\varepsilon^2} \tag{26}$$

*for some large enough universal constant $c_{\mathsf{k}} > 0$. With probability at least $1 - \delta$, the product policy $\widehat{\pi}_1 \times \widehat{\pi}_2$ computed by Algorithm 1 is an $\varepsilon$-approximate Nash equilibrium, i.e., its sub-optimality gap (cf. (13)) obeys $\mathtt{gap}\big(\widehat{\pi}_1 \times \widehat{\pi}_2\big) \le \varepsilon$.*

**Theorem 2** (CCE for multi-player general-sum MGs)**.** *Consider an $m$-player general-sum Markov game, and consider any $\varepsilon \in (0, H]$ and any $0 < \delta < 1$. Suppose that*

$$K \ge \frac{c_{\mathsf{k}} H^3 \log^4 \big( \frac{KS \sum_{i=1}^m A_i}{\delta} \big)}{\varepsilon^2} \tag{27}$$

*for some large enough universal constant $c_{\mathsf{k}} > 0$. With probability at least $1 - \delta$, the joint policy $\widehat{\pi}$ returned by Algorithm 1 is an $\varepsilon$-approximate CCE, i.e., its sub-optimality gap (cf. (13)) obeys $\mathtt{gap}\big(\widehat{\pi}\big) \le \varepsilon$.*

Theorems 1-2 establish sample complexity upper bounds for the proposed algorithm, which we take a moment to interpret as follows. The proofs of these two theorems are postponed to Appendix C in the supplementary material.

**Sample complexity.** When a generative model is available, Theorems 1-2 assert that the total number of samples (i.e., $KSH \sum_i A_i$) needed for Algorithm 1 to work is

$$\begin{cases} \widetilde{O}\big( \frac{H^4 S(A_1 + A_2)}{\varepsilon^2} \big), & \text{for learning an } \varepsilon\text{-NE in two-player zero-sum MGs;} \\ \widetilde{O}\big( \frac{H^4 S \sum_{i=1}^m A_i}{\varepsilon^2} \big), & \text{for learning an } \varepsilon\text{-CCE in multi-player general-sum MGs.} \end{cases} \tag{28}$$

As far as we know, our theorems deliver the first results that uncover the plausibility of simultaneously overcoming the long-horizon barrier and the curse of multi-agents. Let us compare (28) with prior art.

- *NE in two-player zero-sum MGs.* First, consider learning $\varepsilon$-NE policies in two-player zero-sum MGs. In comparison to Zhang et al. [79] (cf. (1)), our result reveals that what ultimately matters is the total number of individual actions (i.e., $A_1 + A_2$) as opposed to the total number $A_1 A_2$ of possible joint actions; additionally, our results exhibit improved horizon dependency (by a factor of $H^2$) compared to Bai et al. [5], Jin et al. [31] (see (3)), although we remark that the online sampling protocol therein is clearly more restrictive than a generative model.

- *CCE in multi-player general-sum MGs (for a fixed $m$).* Similar messages carry over to the task of learning multi-player general-sum MGs when the number of players $m$ is a fixed constant. Liu et al. [43] provided the first non-asymptotic result on learning CCE in the exploration setting; the model-based algorithm studied therein learns an $\varepsilon$-CCE using

$$\widetilde{O}\bigg( \frac{H^5 S^2 \prod_{i=1}^m A_i}{\varepsilon^2} \bigg) \text{ samples} \qquad \text{or} \qquad \widetilde{O}\bigg( \frac{H^4 S^2 \prod_{i=1}^m A_i}{\varepsilon^2} \bigg) \text{ episodes} \tag{29}$$

which is sub-optimal in terms of the dependency on both $H$ and $S$ and suffers from the curse of multiple agents. A more recent strand of works focused on a type of online RL algorithms called V-learning, which exploited the effectiveness of adversarial learning subroutines in overcoming the curse of multi-agents [44, 63, 31]; along this line, the state-of-the-art sample complexity bound is [31]:

$$\widetilde{O}\left(\frac{H^6 S \max_{1 \leq i \leq m} A_i}{\varepsilon^2}\right) \text{ samples} \qquad \text{or} \qquad \widetilde{O}\left(\frac{H^5 S \max_{1 \leq i \leq m} A_i}{\varepsilon^2}\right) \text{ episodes,} \qquad (30)$$

which remains suboptimal in terms of the horizon dependency. As a drawback of these works, the policy returned by V-learning is non-Markovian, an issue that has been recently addressed by Daskalakis et al. [23] at the price of a much higher sample complexity. It is worth emphasizing that all these works assume the online exploration setting as opposed to the scenario with a generative model.

**Minimax optimality.** To assess the tightness of our result (28), it is helpful to look at the information-theoretic limit. Following the minimax lower bound for single-agent MDPs [3, 41], one can develop a minimax sample complexity lower bound for Markov games (w.r.t. finding either an $\varepsilon$-NE or an $\varepsilon$-CCE) that scales as

$$\text{(minimax lower bound)} \qquad \frac{H^4 S \max_{1 \leq i \leq m} A_i}{\varepsilon^2} \qquad (31)$$

modulo some logarithmic factor; see Appendix E.3 in the supplementary material for a formal statement and its proof. Taking this together with (28) confirms the minimax optimality of our algorithm (up to logarithmic terms) when the number $m$ of players is fixed or grows only logarithmically in problem parameters.

**No burn-in sample size and full $\varepsilon$-range.** It is noteworthy that the validity of our sample complexity bound (28) is guaranteed for the entire range of $\varepsilon$-levels (i.e., any $\varepsilon \in (0, H]$). This feature is particularly appealing in the data-starved applications, as it implies that there is no burn-in sample size needed for our algorithm to work optimally.

**Miscellaneous properties of our algorithm.** Finally, we would like to remark in passing that our learning algorithm enjoys several properties that might be practically appealing. For instance, the output policies are Markovian in nature, which depend only on the current state $s$ and step number $h$. This is enabled thanks to the availability of the generative model, which allows us to settle the sampling and learning process for step $h + 1$ completely before moving backward to step $h$; in contrast, the online sampling protocol studied in Bai et al. [5], Jin et al. [31] cannot be implemented in this way without incurring information loss. In addition, our algorithm can be carried out in a decentralized fashion (except that the final estimate $\widehat{\pi}$ needs to aggregate policy iterates from all players), with each player acting in a symmetric yet independent manner (without the need of knowing each other's individual action). Our algorithm is also "rational" in the sense that it converges to the best-response policy of a player if all other players freeze their policies. All this is achieved under minimal sample complexity with the aid of the generative model.

## 4 Discussion

The primary contribution of this paper has been to develop a sample-optimal paradigm that simultaneously overcomes the curse of multiple agents and optimizes the horizon dependency when solving multi-player Markov games. This goal was not accomplished in any of the previous works, regardless of the sampling mechanism in use. The adoption of the adversarial learning subroutine helps break the curse of multiple agents compared to the prior model-based approach [79, 43], whereas the availability of the generative model in conjunction with the variance-aware bonus design improves horizon dependency compared to Bai et al. [5], Jin et al. [31].

**Acknowledgements:** Y. Chen is supported in part by the Alfred P. Sloan Research Fellowship, the Google Research Scholar Award, the AFOSR grant FA9550-22-1-0198, the ONR grant N00014-22-1-2354, and the NSF grants CCF-2221009, CCF-1907661, IIS-2218713 and IIS-2218773. Y. Wei is supported in part by the the NSF grants CCF-2106778, DMS-2147546/2015447 and CAREER award DMS-2143215. Y. Chi are supported in part by the grants ONR N00014-19-1-2404, NSF CCF-2106778 and DMS-2134080, and CAREER award ECCS-1818571. Part of this work was done while G. Li, Y. Wei and Y. Chen were visiting the Simons Institute for the Theory of Computing.

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
