# A    Other related works

Let us discuss in passing additional prior works on learning equilibrium solutions in MARL, which have attracted an explosion of interest in recent years. While the Nash equilibrium is arguably the most compelling solution concept in Markov games, the finite-sample/finite-time studies of NE learning concentrate primarily on two-player zero-sum MGs (e.g., Bai and Jin [4], Chen et al. [16], Mao and Başar [44], Wei et al. [70], Tian et al. [64], Cui and Du [17, 18], Zhong et al. [83], Jia et al. [29], Yang and Ma [76], Yan et al. [74], Dou et al. [24]), mainly because computing NEs becomes, for the most part, computationally infeasible (i.e., PPAD-complete) when going beyond two-player zero-sum MGs [20, 21]. Roughly speaking, previous NE-finding algorithms for two-player zero-sum Markov games can be categorized into model-based algorithms [52, 79, 43], value-based algorithms [4, 5, 73, 54, 31, 15], and policy-based algorithms [10, 22, 71, 82, 14, 81, 11]. In particular, Bai et al. [5], Jin et al. [31] developed the first algorithms to beat the curse of multiple agents in two-player zero-sum MGs, while Jin et al. [31], Daskalakis et al. [23], Mao and Başar [44], Song et al. [63] further demonstrated how to accomplish the same goal when learning other computationally tractable solution concepts (e.g., coarse correlated equilibria) in general-sum multi-player Markov games. The recent works Cui and Du [17, 18], Yan et al. [74] studied how to alleviate the sample size scaling with the number of agents in the presence of offline data, with Cui and Du [18] providing a sample-efficient algorithm that also learns NEs in multi-agent Markov games (despite computational intractability).

We shall also briefly remark on the prior works that concern RL with a generative model. While there are multiple sampling mechanisms (e.g., online exploratory sampling, offline data) that bear practical relevance, the generative model (or simulator) serves as an idealistic sampling protocol that has received much recent attention, covering the design of various model-based, model-free and policy-based algorithms [35, 69, 1, 3, 38, 32, 60, 67, 68, 39, 34, 61, 51, 36, 13, 26, 7, 71, 65, 77, 72, 46, 75, 78, 25]. In single-agent RL, the model-based approach has been shown to be minimax-optimal for the entire $\varepsilon$-range [38, 1, 3]. When it comes to multi-agent RL, sample-efficient solutions with a generative model have been proposed in the recent works [62, 19, 79], although a provably sample-optimal strategy was previously unavailable.

# B    Regret bounds for FTRL via variance-type quantities

Before embarking on our analysis for Markov games, we take a detour to study the celebrated Follow-the-Regularized-Leader algorithm for online weighted linear optimization, which plays a central role in the analysis of Markov games.

## B.1    Setting: online learning for weighted linear optimization

Let $\ell_1, \ldots, \ell_n \in \mathbb{R}^A$ represent an arbitrary sequence of *non-negative* loss vectors. We focus on the following setting of online learning or adversarial learning [37]: in each round $k$,

1. the learner makes a randomized prediction by choosing a distribution $\pi_k \in \Delta(\mathcal{A})$ over the actions in $\mathcal{A} = \{1, \cdots, A\}$;
2. subsequently, the learner observes the loss vector $\ell_k$, which is permitted to be adversarially chosen.

To evaluate the performance of the learner, we resort to a regret metric w.r.t. a certain weighted linear objective function. To be precise, consider a non-negative sequence $\{\alpha_k\}_{1 \le k \le n}$ with $0 \le \alpha_k \le 1$; for each $1 \le k \le n$, we define recursively the following weighted average of the loss vectors:

$$L_0 = 0 \qquad \text{and} \qquad L_k = (1 - \alpha_k)L_{k-1} + \alpha_k \ell_k, \quad k \ge 1,$$

which can be easily shown to enjoy the following expression

$$L_k = \sum_{i=1}^{k} \alpha_i^k \ell_k$$

with $\alpha_i^k$ defined in (5). When the sequential predictions made by the learner are $\{\pi_k\}_{k \ge 1}$, we define the associated regret w.r.t. the above weighted sum of loss vectors as follows:

$$R_n := \max_{a \in \mathcal{A}} R_n(a) \qquad \text{with } R_n(a) := \sum_{k=1}^{n} \alpha_k^n \langle \pi_k, \ell_k \rangle - \sum_{k=1}^{n} \alpha_k^n \ell_k(a), \tag{32}$$

which compares the learner's performance (i.e., the expected loss of the learner over time if it draws actions based on $\pi_k$ in round $k$) against that of the best *fixed* action in hindsight.

## B.2 Refined regret bounds for FTRL

**Follow-the-Regularized-Leader.** The FTRL algorithm [57, 55] tailored to the above online optimization setting adopts the following update rule:

$$\pi_{k+1} = \arg \min_{\pi \in \Delta(\mathcal{A})} \left\{ \langle \pi, L_k \rangle + \frac{1}{\eta_{k+1}} F(\pi) \right\}, \qquad k = 1, 2, \ldots \tag{33}$$

where $\eta_{k+1} > 0$ denotes the learning rate, and $F(\cdot)$ is some convex regularization function employed to stabilize the learning process [56]. Throughout this section, we restrict our attention to negative-entropy regularization, namely,

$$F(\pi) = \sum_{a \in \mathcal{A}} \pi(a) \log \big( \pi(a) \big),$$

which allows one to express the FTRL update rule as the following exponential weights strategy (see, e.g., Lattimore and Szepesvári [37, Section 28.1])

$$\pi_{k+1}(a) = \frac{\exp \big( -\eta_{k+1} L_k(a) \big)}{\sum_{a' \in \mathcal{A}} \exp \big( -\eta_{k+1} L_k(a') \big)} \qquad \text{for all } a \in \mathcal{A}. \tag{34}$$

This update rule is also intimately connected to online mirror descent [37].

**Refined regret bounds via variance-style quantities.** As it turns out, the regret of FTRL can be upper bounded by certain (weighted) variance-type quantities, as asserted by the following theorem.

**Theorem 3.** *Suppose that $0 < \alpha_1 \leq 1$ and $\eta_1 = \eta_2(1 - \alpha_1)$. Also, assume that $0 < \alpha_k < 1$ and $0 < \eta_{k+1}(1 - \alpha_k) \leq \eta_k$ for all $k \geq 2$. In addition, define*

$$\widehat{\eta}_k := \begin{cases} \eta_2, & \text{if } k = 1, \\ \frac{\eta_k}{1 - \alpha_k}, & \text{if } k > 1. \end{cases} \tag{35}$$

*Then the regret (cf. (32)) of the FTRL algorithm satisfies*

$$R_n \leq \frac{5}{3} \sum_{k=1}^{n} \alpha_k^n \widehat{\eta}_k \alpha_k \mathsf{Var}_{\pi_k}(\ell_k) + \frac{\log A}{\eta_{n+1}} + 3 \sum_{k=1}^{n} \alpha_k^n \widehat{\eta}_k^2 \alpha_k^2 \|\ell_k\|_\infty^3 \, \mathbb{1}\left( \widehat{\eta}_k \alpha_k \|\ell_k\|_\infty > \frac{1}{3} \right), \tag{36}$$

*where for any $\ell \in \mathbb{R}^A$ and any $\pi \in \Delta(\mathcal{A})$ we define*

$$\mathsf{Var}_\pi(\ell) := \sum_a \pi(a) \Big( \ell(a) - \sum_{a'} \pi(a')\ell(a') \Big)^2.$$

***Remark** 1.* Note that the FTRL algorithm and the data generating process in this section are both described in a completely deterministic manner; no randomness is involved in the above theorem even though we introduce the variance-style quantities.

The proof of Theorem 3 is postponed to Appendix D. Let us take a moment to discuss the key distinction between Theorem 3 and prior theory.

- A key term in the regret bound (36) is a weighted sum of the "variance-style" quantities $\{\mathsf{Var}_{\pi_k}(\ell_k)\}$. In comparison, prior regret bounds typically involve the norm-type quantities (e.g., the infinity norms $\{\|\ell_k\|_\infty^2\}$) as opposed to the "variances"; see, for instance, Lattimore and Szepesvári [37, Corollary 28.8] for a representative existing regret bound that takes the form of the sum of $\{\|\ell_k\|_\infty^2\}$ that takes the form of the sum of $\{\|\ell_k\|_\infty^2\}$.[2] While $\mathsf{Var}(\ell_k) \leq \|\ell_k\|_\infty^2$ is orderwise tight in the worst-case scenario for a given iteration $k$, exploiting the problem-specific variance-type structure across time is crucial in sharpening the horizon dependence in many RL problems (e.g., Azar et al. [3], Jin et al. [30], Li et al. [41, 40]).

---

[2]Note that the Bregman divergence generated by the negative entropy function is the (generalized) KL divergence [6], which is strongly convex w.r.t. $\|\cdot\|_1$ due to Pinsker's inequality. Additionally, the dual norm of $\|\cdot\|_1$ is the infinity norm.

- The careful reader would remark that the final term of (36) relies on the infinity norm $\|\ell_k\|_\infty$ as well. Fortunately, when the products of the learning rates $\widehat{\eta}_k \alpha_k$ are chosen to be diminishing (which is the case in our analysis for Markov games), the number of iterations obeying $\widehat{\eta}_k \alpha_k \|\ell_k\|_\infty > 1/3$ is reasonably small, thus ensuring that this term does not exert too much of an influence on the regret bound.

## C  Proof of Theorems 1-2

To begin with, we claim that Theorem 1 is a direct consequence of Theorem 2. Towards this, note that in a two-player zero-sum Markov game, it is self-evident that $\widehat{\pi}_{-1} = \widehat{\pi}_2$ and $\widehat{\pi}_{-2} = \widehat{\pi}_1$ (see line 12 of Algorithm 1). Consequently, Theorem 2 (if it is valid) reveals that

$$\varepsilon \geq \mathsf{gap}(\widehat{\pi}; s) = \max\left\{ V_{1,1}^{\star,\widehat{\pi}_{-1}}(s) - V_{1,1}^{\widehat{\pi}}(s),\ V_{2,1}^{\star,\widehat{\pi}_{-2}}(s) - V_{2,1}^{\widehat{\pi}}(s) \right\}$$
$$= \max\left\{ V_{1,1}^{\star,\widehat{\pi}_2}(s) - V_{1,1}^{\widehat{\pi}}(s),\ V_{2,1}^{\widehat{\pi}_1,\star}(s) - V_{2,1}^{\widehat{\pi}}(s) \right\}, \qquad \text{for all } s \in \mathcal{S}. \quad (37)$$

Moreover, recalling that $r_{1,h} = -r_{2,h}$ for all $h \in [H]$, one has $V_{1,1}^{\pi}(s) = -V_{2,1}^{\pi}(s)$ for any joint policy profile $\pi$, which taken collectively with (37) results in

$$V_{1,1}^{\star,\widehat{\pi}_2}(s) - V_{1,1}^{\widehat{\pi}_1 \times \widehat{\pi}_2}(s) = V_{1,1}^{\star,\widehat{\pi}_2}(s) + V_{2,1}^{\widehat{\pi}_1 \times \widehat{\pi}_2}(s) \leq V_{1,1}^{\star,\widehat{\pi}_2}(s) + V_{2,1}^{\widehat{\pi}_1,\star}(s)$$
$$= V_{1,1}^{\star,\widehat{\pi}_2}(s) - V_{1,1}^{\widehat{\pi}}(s) + V_{2,1}^{\widehat{\pi}_1,\star}(s) - V_{2,1}^{\widehat{\pi}}(s) \leq 2\varepsilon.$$

Analogously, one has $V_{2,1}^{\widehat{\pi}_1,\star}(s) - V_{2,1}^{\widehat{\pi}_1 \times \widehat{\pi}_2}(s) \leq 2\varepsilon$. Replacing $\varepsilon$ with $\varepsilon/2$ immediately establishes Theorem 1.

With the above argument in mind, the remainder of this section is devoted to proving Theorem 2.

### C.1  Preliminaries and notation

Let us start with some preliminary facts and notation. Given that $\varepsilon \leq H$, the assumption (27) requires

$$K \geq c_{\mathsf{k}} H \log^4\left( \frac{KS \sum_i A_i}{\delta} \right) \tag{38}$$

for some large enough constant $c_{\mathsf{k}} > 0$, which will be a condition assumed throughout the proof. We also gather below several basic facts about our choices of learning rates $\{\alpha_i\}$ (cf. (22)) and the corresponding quantities $\{\alpha_i^k\}$ (cf. (5)).

**Lemma 1.** *For any $k \geq 1$, one has*

$$\alpha_1 = 1, \qquad \sum_{i=1}^{k} \alpha_i^k = 1, \qquad \max_{1 \leq i \leq k} \alpha_i^k \leq \frac{2c_\alpha \log K}{k}. \tag{39a}$$

*In addition, if $k \geq c_\alpha \log K + 1$ and $c_\alpha \geq 24$, then one has*

$$\max_{1 \leq i \leq k/2} \alpha_i^k \leq 1/K^6. \tag{39b}$$

*Proof.* The result (39a) is standard and has been recorded in previous works (e.g., Jin et al. [30, Appendix B]). Regarding (39b), we note that for any $i \leq k/2$ and $k \geq c_\alpha \log K + 1$,

$$\alpha_i^k \leq \prod_{j=i+1}^{k} (1-\alpha_j) \leq \prod_{j=k/2+1}^{k} (1-\alpha_j) \leq (1-\alpha_k)^{k/2} \leq \left(1 - \frac{c_\alpha \log K}{2k}\right)^{k/2} \leq \exp\left(-\frac{c_\alpha \log K}{4}\right) \leq \frac{1}{K^6},$$

where we have used the fact that $\alpha_k = \frac{c_\alpha \log K}{k-1+c_\alpha \log K} \geq \frac{c_\alpha \log K}{2k}$ and the assumption $c_\alpha \geq 24$. $\qquad\square$

Additionally, recognizing the definition in (15) and the upper bound $\widehat{V}_{i,h+1}(s) \leq H - h$ (cf. (20a)), we make note of the range of the iterates $\{q_{i,h}^k\}$ as follows.

**Lemma 2.** *For any $i \in [m]$ and any $(h, k, s, a_i) \in [H] \times [K] \times \mathcal{S} \times \mathcal{A}_i$, it holds that*

$$0 \le q_{i,h}^k(s, a_i) \le H - h + 1. \tag{40}$$

Next, we introduce several additional notation that helps simplify our presentation of the proof. For any policy $\mu : \mathcal{S} \times [H] \to \Delta(\mathcal{A}_i)$, we adopt the convenient notation

$$\mu_h(s) := \mu_h(\cdot \,|\, s) \in \Delta(\mathcal{A}_i).$$

We shall also employ the expectation operator $\mathbb{E}_{h,k-1}[\cdot]$ (resp. variance operator $\mathsf{Var}_{h,k-1}[\cdot]$) to denote the expectation (resp. variance) conditional on what happens before the beginning of the $k$-th round of data collection for step $h$ (see Section 3.1 about the data collection process).

## C.2  Proof outline

With the above preliminaries in place, we are in a position to present our analysis. Recall that the joint policy $\widehat{\pi}$ computed by Algorithm 1 takes the form of a mixture of product policies

$$\sum_{k=1}^{K} \alpha_k^K \underbrace{\left( \pi_{1,h}^k \times \cdots \times \pi_{m,h}^k \right)}_{=: \pi_h^k} \tag{41}$$

at step $h$. Consequently, the value function under policy $\widehat{\pi}$ satisfies the following Bellman equation:

$$V_{i,H+1}^{\widehat{\pi}}(s) := 0 \tag{42a}$$

$$V_{i,h}^{\widehat{\pi}}(s) := \sum_{k=1}^{K} \sum_{\boldsymbol{a} \in \mathcal{A}} \alpha_k^K \pi_h^k(\boldsymbol{a} \,|\, s) \Big[ r_{i,h}(s, \boldsymbol{a}) + \big\langle P_h(\cdot \,|\, s, \boldsymbol{a}), V_{i,h+1}^{\widehat{\pi}} \big\rangle \Big] \tag{42b}$$

for all $(i, s, h) \in [m] \times \mathcal{S} \times [H]$. To establish Theorem 2, we seek to prove the following inequality:

$$V_{i,1}^{\star, \widehat{\pi}_{-i}}(s) - V_{i,1}^{\widehat{\pi}}(s) \le \varepsilon, \qquad 1 \le i \le m, \tag{43}$$

where we remind the reader of the definition of $V_{i,1}^{\star, \widehat{\pi}_{-i}}$ in (9).

Towards this, let us introduce the following best-response policy of the $i$-th player:

$$\widetilde{\pi}_i^\star = \big[ \widetilde{\pi}_{i,h}^\star \big]_{h \in [H]} := \arg \max_{\pi_i' : \mathcal{S} \times [H] \to \Delta(\mathcal{A}_i)} V_{i,1}^{\pi_i' \times \widehat{\pi}_{-i}}.$$

We make note of the following key decomposition:

$$V_{i,h}^{\star, \widehat{\pi}_{-i}} - V_{i,h}^{\widehat{\pi}} \le \big( V_{i,h}^{\star, \widehat{\pi}_{-i}} - \overline{V}_{i,h}^{\widetilde{\pi}_i^\star \times \widehat{\pi}_{-i}} \big) + \big( \overline{V}_{i,h}^{\star, \widehat{\pi}_{-i}} - \overline{V}_{i,h}^{\widehat{\pi}} \big) + \big( \overline{V}_{i,h}^{\widehat{\pi}} - V_{i,h}^{\widehat{\pi}} \big), \tag{44}$$

where we define the following auxiliary value functions:

$$\overline{V}_{i,h}^{\widetilde{\pi}_i^\star \times \widehat{\pi}_{-i}}(s) := \sum_{k=1}^{K} \alpha_k^K \mathop{\mathbb{E}}_{a_i \sim \widetilde{\pi}_{i,h}^\star(s)} \Big[ r_{i,h}^k(s, a_i) + \big\langle P_{i,h}^k(\cdot \,|\, s, a_i), \overline{V}_{i,h+1}^{\widetilde{\pi}_i^\star \times \widehat{\pi}_{-i}} \big\rangle \Big], \qquad \text{with } \overline{V}_{i,H+1}^{\widetilde{\pi}_i^\star \times \widehat{\pi}_{-i}} = 0,$$

$$\tag{45a}$$

$$\overline{V}_{i,h}^{\star, \widehat{\pi}_{-i}}(s) := \max_{a_i \in \mathcal{A}_i} \sum_{k=1}^{K} \alpha_k^K \Big[ r_{i,h}^k(s, a_i) + \big\langle P_{i,h}^k(\cdot \,|\, s, a_i), \overline{V}_{i,h+1}^{\star, \widehat{\pi}_{-i}} \big\rangle \Big], \qquad \text{with } \overline{V}_{i,H+1}^{\star, \widehat{\pi}_{-i}} = 0,$$

$$\tag{45b}$$

$$\overline{V}_{i,h}^{\widehat{\pi}}(s) := \sum_{k=1}^{K} \alpha_k^K \mathop{\mathbb{E}}_{a_i \sim \pi_{i,h}^k(s)} \Big[ r_{i,h}^k(s, a_i) + \big\langle P_{i,h}^k(\cdot \,|\, s, a_i), \overline{V}_{i,h+1}^{\widehat{\pi}} \big\rangle \Big], \qquad \text{with } \overline{V}_{i,H+1}^{\widehat{\pi}} = 0.$$

$$\tag{45c}$$

Here, we have used the elementary fact $\overline{V}_{i,h}^{\widetilde{\pi}_i^\star \times \widehat{\pi}_{-i}} \le \overline{V}_{i,h}^{\star, \widehat{\pi}_{-i}}$. We shall establish bounds for the above terms in (44), which consists of three steps as outlined below.

**Step 1: showing that $\widehat{V}_{i,h}$ is an entrywise upper bound on $\overline{V}_{i,h}^{\star,\widehat{\pi}-i}$.** The following lemma ascertains that the value estimate $\widehat{V}_{i,h}$ of the $i$-th player returned by Algorithm 1 is an optimistic estimate of the auxiliary value $\overline{V}_{i,h}^{\star,\widehat{\pi}-i}$ defined in (45b). Evidently, this result cannot happen unless the bonus terms are suitably chosen.

**Lemma 3.** *With probability at least $1 - \delta$, it holds that*

$$\widehat{V}_{i,h} \geq \overline{V}_{i,h}^{\star,\widehat{\pi}-i}, \qquad \text{for all } (i,h) \in [m] \times [H]. \tag{46}$$

The proof of this lemma is postponed to Appendix E.1. Armed with Lemma 3, we can further bound (44) as follows

$$V_{i,h}^{\star,\widehat{\pi}-i} - V_{i,h}^{\widehat{\pi}} \leq \left( V_{i,h}^{\star,\widehat{\pi}-i} - \overline{V}_{i,h}^{\widetilde{\pi}_i^\star \times \widehat{\pi}-i} \right) + \left( \widehat{V}_{i,h} - \overline{V}_{i,h}^{\widehat{\pi}} \right) + \left( \overline{V}_{i,h}^{\widehat{\pi}} - V_{i,h}^{\widehat{\pi}} \right). \tag{47}$$

**Step 2: establishing a key recursion.** Recall the definition of $\pi_h^k$ in (41). Let us define the following auxiliary reward vectors $r_{i,h}^{\widehat{\pi}}, r_{i,h}^{\widetilde{\pi}_i^\star \times \widehat{\pi}-i}, \overline{r}_{i,h} \in \mathbb{R}^S$ as well as the auxiliary probability transition matrices $P_{i,h}^{\widehat{\pi}}, P_{i,h}^{\widetilde{\pi}_i^\star \times \widehat{\pi}-i}, \overline{P}_{i,h} \in \mathbb{R}^{S \times S}$ such that: for any $s, s' \in \mathcal{S}$,

$$r_{i,h}^{\widehat{\pi}}(s) := \sum_{k=1}^K \alpha_k^K \mathop{\mathbb{E}}_{\boldsymbol{a} \sim \pi_h^k(s)} \left[ r_{i,h}(s, \boldsymbol{a}) \right], \tag{48a}$$

$$P_{i,h}^{\widehat{\pi}}(s, s') := \sum_{k=1}^K \alpha_k^K \mathop{\mathbb{E}}_{\boldsymbol{a} \sim \pi_h^k(s)} \left[ P_h(s' \,|\, s, \boldsymbol{a}) \right], \tag{48b}$$

$$r_{i,h}^{\widetilde{\pi}_i^\star \times \widehat{\pi}-i}(s) := \sum_{k=1}^K \alpha_k^K \mathop{\mathbb{E}}_{(a_i, \boldsymbol{a}-i) \sim \widetilde{\pi}_{i,h}^\star(s) \times \pi_{-i,h}^k(s)} \left[ r_{i,h}(s, \boldsymbol{a}) \right], \tag{48c}$$

$$P_{i,h}^{\widetilde{\pi}_i^\star \times \widehat{\pi}-i}(s, s') := \sum_{k=1}^K \alpha_k^K \mathop{\mathbb{E}}_{(a_i, \boldsymbol{a}-i) \sim \widetilde{\pi}_{i,h}^\star(s) \times \pi_{-i,h}^k(s)} \left[ P_h(s' \,|\, s, \boldsymbol{a}) \right], \tag{48d}$$

$$\overline{r}_{i,h}(s) := \sum_{k=1}^K \alpha_k^K \sum_{a_i \in \mathcal{A}_i} \pi_{i,h}^k(a_i \,|\, s) r_{i,h}^k(s, a_i), \tag{48e}$$

$$\overline{P}_{i,h}(s, s') := \sum_{k=1}^K \alpha_k^K \sum_{a_i \in \mathcal{A}_i} \pi_{i,h}^k(a_i \,|\, s) P_{i,h}^k(s' \,|\, s, a_i). \tag{48f}$$

As it turns out, $\overline{V}_{i,h}^{\widehat{\pi}}$ (resp. $\overline{V}_{i,h}^{\widetilde{\pi}_i^\star \times \widehat{\pi}-i}$, $\widehat{V}_{i,h}$) stays reasonably close to the "one-step-look-ahead" expression $r_{i,h}^{\widehat{\pi}} + P_{i,h}^{\widehat{\pi}} \overline{V}_{i,h+1}^{\widehat{\pi}}$ (resp. $r_{i,h}^{\widetilde{\pi}_i^\star \times \widehat{\pi}-i} + P_{i,h}^{\widetilde{\pi}_i^\star \times \widehat{\pi}-i} \overline{V}_{i,h+1}^{\widetilde{\pi}_i^\star \times \widehat{\pi}-i}$, $\overline{r}_{i,h} + \overline{P}_{i,h} \widehat{V}_{i,h+1}$), as revealed by the recursive relations stated in the following lemma; the proof of this lemma is deferred to Appendix E.2.

**Lemma 4.** *There exists some universal constant $c_3 > 0$ such that with probability exceeding $1 - \delta$,*

$$\left| \overline{V}_{i,h}^{\widehat{\pi}} - \left( r_{i,h}^{\widehat{\pi}} + P_{i,h}^{\widehat{\pi}} \overline{V}_{i,h+1}^{\widehat{\pi}} \right) \right| \leq c_3 \sqrt{\frac{H \log^3 \left( \frac{KS \sum_i A_i}{\delta} \right)}{K}} \mathbf{1}$$

$$+ c_3 \sqrt{\frac{\log^3 \left( \frac{KS \sum_i A_i}{\delta} \right)}{KH}} \left[ P_{i,h}^{\widehat{\pi}} \left( \overline{V}_{i,h+1}^{\widehat{\pi}} \circ \overline{V}_{i,h+1}^{\widehat{\pi}} \right) - \left( P_{i,h}^{\widehat{\pi}} \overline{V}_{i,h+1}^{\widehat{\pi}} \right) \circ \left( P_{i,h}^{\widehat{\pi}} \overline{V}_{i,h+1}^{\widehat{\pi}} \right) \right], \tag{49a}$$

$$\left| \overline{V}_{i,h}^{\widetilde{\pi}_i^\star \times \widehat{\pi}-i} - \left( r_{i,h}^{\widetilde{\pi}_i^\star \times \widehat{\pi}-i} + P_{i,h}^{\widetilde{\pi}_i^\star \times \widehat{\pi}-i} \overline{V}_{i,h+1}^{\widetilde{\pi}_i^\star \times \widehat{\pi}-i} \right) \right| \leq c_3 \sqrt{\frac{H \log^3 \left( \frac{KS \sum_i A_i}{\delta} \right)}{K}} \mathbf{1}$$

$$+ c_3 \sqrt{\frac{\log^3 \left( \frac{KS \sum_i A_i}{\delta} \right)}{KH}} \left[ P_{i,h}^{\widetilde{\pi}_i^\star \times \widehat{\pi}-i} \left( \overline{V}_{i,h+1}^{\widetilde{\pi}_i^\star \times \widehat{\pi}-i} \circ \overline{V}_{i,h+1}^{\widetilde{\pi}_i^\star \times \widehat{\pi}-i} \right) - \left( P_{i,h}^{\widetilde{\pi}_i^\star \times \widehat{\pi}-i} \overline{V}_{i,h+1}^{\widetilde{\pi}_i^\star \times \widehat{\pi}-i} \right) \circ \left( P_{i,h}^{\widetilde{\pi}_i^\star \times \widehat{\pi}-i} \overline{V}_{i,h+1}^{\widetilde{\pi}_i^\star \times \widehat{\pi}-i} \right) \right], \tag{49b}$$

$$\left| \widehat{V}_{i,h} - \left( \overline{r}_{i,h} + \overline{P}_{i,h} \widehat{V}_{i,h+1} \right) \right| \le c_3 \sqrt{\frac{H \log^3 \left( \frac{KS \sum_i A_i}{\delta} \right)}{K}} \mathbf{1}$$

$$+ c_3 \sqrt{\frac{\log^3 \left( \frac{KS \sum_i A_i}{\delta} \right)}{KH}} \left[ \overline{P}_{i,h} \left( \widehat{V}_{i,h+1} \circ \widehat{V}_{i,h+1} \right) - \left( \overline{P}_{i,h} \widehat{V}_{i,h+1} \right) \circ \left( \overline{P}_{i,h} \widehat{V}_{i,h+1} \right) \right]$$

(49c)

*hold for all $h \in [H]$.*

**Remark** 2. The right-hand side of each of the bounds in (49) contains a variance-style term (e.g., those terms taking the form of $P_{i,h}(V_{i,h+1} \circ V_{i,h+1}) - (P_{i,h}V_{i,h+1}) \circ (P_{i,h}V_{i,h+1})$ for some probability transition matrix $P_{i,h}$ and value vector $V_{i,h+1}$). Such variance-style terms are direct consequences of our Bernstein-style bonus terms, and are crucial in optimizing the horizon dependency.

With the above lemma in place, one can readily show that

$$\left| \overline{V}_{i,h}^{\widehat{\pi}} - P_{i,h}^{\widehat{\pi}} \overline{V}_{i,h+1}^{\widehat{\pi}} \right| \le r_{i,h}^{\widehat{\pi}} + c_3 \sqrt{\frac{H \log^3 \left( \frac{KS \sum_i A_i}{\delta} \right)}{K}} \mathbf{1}$$

$$+ \frac{c_3}{H} \sqrt{\frac{H \log^3 \left( \frac{KS \sum_i A_i}{\delta} \right)}{K}} \left[ P_{i,h}^{\widehat{\pi}} \left( \overline{V}_{i,h+1}^{\widehat{\pi}} \circ \overline{V}_{i,h+1}^{\widehat{\pi}} \right) - \left( P_{i,h}^{\widehat{\pi}} \overline{V}_{i,h+1}^{\widehat{\pi}} \right) \circ \left( P_{i,h}^{\widehat{\pi}} \overline{V}_{i,h+1}^{\widehat{\pi}} \right) \right]$$

$$\le \frac{c_4}{4} \mathbf{1} + \frac{1}{4H} \left[ P_{i,h}^{\widehat{\pi}} \left( \overline{V}_{i,h+1}^{\widehat{\pi}} \circ \overline{V}_{i,h+1}^{\widehat{\pi}} \right) - \left( P_{i,h}^{\widehat{\pi}} \overline{V}_{i,h+1}^{\widehat{\pi}} \right) \circ \left( P_{i,h}^{\widehat{\pi}} \overline{V}_{i,h+1}^{\widehat{\pi}} \right) \right] =: \zeta_0$$

(50)

for some large enough constant $c_4 > 0$, where the last line holds due to Condition (38), the basic fact $P_{i,h}^{\widehat{\pi}} \left( \overline{V}_{i,h+1}^{\widehat{\pi}} \circ \overline{V}_{i,h+1}^{\widehat{\pi}} \right) \ge \left( P_{i,h}^{\widehat{\pi}} \overline{V}_{i,h+1}^{\widehat{\pi}} \right) \circ \left( P_{i,h}^{\widehat{\pi}} \overline{V}_{i,h+1}^{\widehat{\pi}} \right)$, and the following fact (for large enough $c_4$)

$$c_3 \sqrt{\frac{H \log^3 \left( \frac{KS \sum_i A_i}{\delta} \right)}{K}} \mathbf{1} + r_{i,h}^{\widehat{\pi}} \le c_3 \sqrt{\frac{H \log^3 \left( \frac{KS \sum_i A_i}{\delta} \right)}{K}} \mathbf{1} + \mathbf{1} \le \frac{c_4}{4} \mathbf{1}.$$

In addition, recalling that $\| \overline{V}_{i,h}^{\widehat{\pi}} \|_\infty, \| \overline{V}_{i,h+1}^{\widehat{\pi}} \|_\infty \le H$ (cf. (20a)) and recognizing that $\zeta_0 \ge 0$ (see (50)), we can demonstrate that

$$\left| \overline{V}_{i,h}^{\widehat{\pi}} \circ \overline{V}_{i,h}^{\widehat{\pi}} - \left( P_{i,h}^{\widehat{\pi}} \overline{V}_{i,h+1}^{\widehat{\pi}} \right) \circ \left( P_{i,h}^{\widehat{\pi}} \overline{V}_{i,h+1}^{\widehat{\pi}} \right) \right| = \left| \left( \overline{V}_{i,h}^{\widehat{\pi}} + P_{i,h}^{\widehat{\pi}} \overline{V}_{i,h+1}^{\widehat{\pi}} \right) \circ \left( \overline{V}_{i,h}^{\widehat{\pi}} - P_{i,h}^{\widehat{\pi}} \overline{V}_{i,h+1}^{\widehat{\pi}} \right) \right|$$

$$\le \left( \overline{V}_{i,h}^{\widehat{\pi}} + P_{i,h}^{\widehat{\pi}} \overline{V}_{i,h+1}^{\widehat{\pi}} \right) \circ \zeta_0 \le 2H \zeta_0$$

$$= \frac{c_4}{2} H \mathbf{1} + \frac{1}{2} \left[ P_{i,h}^{\widehat{\pi}} \left( \overline{V}_{i,h+1}^{\widehat{\pi}} \circ \overline{V}_{i,h+1}^{\widehat{\pi}} \right) - \left( P_{i,h}^{\widehat{\pi}} \overline{V}_{i,h+1}^{\widehat{\pi}} \right) \circ \left( P_{i,h}^{\widehat{\pi}} \overline{V}_{i,h+1}^{\widehat{\pi}} \right) \right].$$

(51)

This further leads to

$$P_{i,h}^{\widehat{\pi}} \left( \overline{V}_{i,h+1}^{\widehat{\pi}} \circ \overline{V}_{i,h+1}^{\widehat{\pi}} \right) - \left( P_{i,h}^{\widehat{\pi}} \overline{V}_{i,h+1}^{\widehat{\pi}} \right) \circ \left( P_{i,h}^{\widehat{\pi}} \overline{V}_{i,h+1}^{\widehat{\pi}} \right)$$

$$= P_{i,h}^{\widehat{\pi}} \left( \overline{V}_{i,h+1}^{\widehat{\pi}} \circ \overline{V}_{i,h+1}^{\widehat{\pi}} \right) - \overline{V}_{i,h}^{\widehat{\pi}} \circ \overline{V}_{i,h}^{\widehat{\pi}} + \overline{V}_{i,h}^{\widehat{\pi}} \circ \overline{V}_{i,h}^{\widehat{\pi}} - \left( P_{i,h}^{\widehat{\pi}} \overline{V}_{i,h+1}^{\widehat{\pi}} \right) \circ \left( P_{i,h}^{\widehat{\pi}} \overline{V}_{i,h+1}^{\widehat{\pi}} \right)$$

$$\le P_{i,h}^{\widehat{\pi}} \left( \overline{V}_{i,h+1}^{\widehat{\pi}} \circ \overline{V}_{i,h+1}^{\widehat{\pi}} \right) - \overline{V}_{i,h}^{\widehat{\pi}} \circ \overline{V}_{i,h}^{\widehat{\pi}} + \frac{c_4}{2} H \mathbf{1} + \frac{1}{2} \left[ P_{i,h}^{\widehat{\pi}} \left( \overline{V}_{i,h+1}^{\widehat{\pi}} \circ \overline{V}_{i,h+1}^{\widehat{\pi}} \right) - \left( P_{i,h}^{\widehat{\pi}} \overline{V}_{i,h+1}^{\widehat{\pi}} \right) \circ \left( P_{i,h}^{\widehat{\pi}} \overline{V}_{i,h+1}^{\widehat{\pi}} \right) \right],$$

which can be rearranged to yield

$$P_{i,h}^{\widehat{\pi}} \left( \overline{V}_{i,h+1}^{\widehat{\pi}} \circ \overline{V}_{i,h+1}^{\widehat{\pi}} \right) - \left( P_{i,h}^{\widehat{\pi}} \overline{V}_{i,h+1}^{\widehat{\pi}} \right) \circ \left( P_{i,h}^{\widehat{\pi}} \overline{V}_{i,h+1}^{\widehat{\pi}} \right) \le 2 \left[ P_{i,h}^{\widehat{\pi}} \left( \overline{V}_{i,h+1}^{\widehat{\pi}} \circ \overline{V}_{i,h+1}^{\widehat{\pi}} \right) - \overline{V}_{i,h}^{\widehat{\pi}} \circ \overline{V}_{i,h}^{\widehat{\pi}} \right] + c_4 H \mathbf{1}.$$

Substituting it into (49a) and combining terms give

$$\left| \overline{V}_{i,h}^{\widehat{\pi}} - \left( r_{i,h}^{\widehat{\pi}} + P_{i,h}^{\widehat{\pi}} \overline{V}_{i,h+1}^{\widehat{\pi}} \right) \right| \le c_5 \sqrt{\frac{H \log^3 \left( \frac{KS \sum_i A_i}{\delta} \right)}{K}} \mathbf{1}$$

$$+ 2c_3 \sqrt{\frac{\log^3 \left( \frac{KS \sum_i A_i}{\delta} \right)}{KH}} \left[ P_{i,h}^{\widehat{\pi}} \left( \overline{V}_{i,h+1}^{\widehat{\pi}} \circ \overline{V}_{i,h+1}^{\widehat{\pi}} \right) - \overline{V}_{i,h}^{\widehat{\pi}} \circ \overline{V}_{i,h}^{\widehat{\pi}} \right],$$

(52)

where we take $c_5 = c_3 + c_3 c_4$.

An analogous argument (which is omitted here for brevity) also reveals that

$$\left| \overline{V}_{i,h}^{\widetilde{\pi}_i^\star \times \widehat{\pi}_{-i}} - \left( r_{i,h}^{\widetilde{\pi}_i^\star \times \widehat{\pi}_{-i}} + P_{i,h}^{\widetilde{\pi}_i^\star \times \widehat{\pi}_{-i}} \overline{V}_{i,h+1}^{\widetilde{\pi}_i^\star \times \widehat{\pi}_{-i}} \right) \right|$$

$$\leq c_5 \sqrt{\frac{H \log^3\left(\frac{KS\sum_i A_i}{\delta}\right)}{K}} 1 + 2c_3 \sqrt{\frac{\log^3\left(\frac{KS\sum_i A_i}{\delta}\right)}{KH}} \left[ P_{i,h}^{\widetilde{\pi}_i^\star \times \widehat{\pi}_{-i}} \left( \overline{V}_{i,h+1}^{\widetilde{\pi}_i^\star \times \widehat{\pi}_{-i}} \circ \overline{V}_{i,h+1}^{\widetilde{\pi}_i^\star, \widehat{\pi}_{-i}} \right) - \overline{V}_{i,h}^{\widetilde{\pi}_i^\star \times \widehat{\pi}_{-i}} \circ \overline{V}_{i,h}^{\widetilde{\pi}_i^\star \times \widehat{\pi}_{-i}} \right],$$

(53)

$$\left| \widehat{V}_{i,h} - \left( \overline{r}_{i,h} + \overline{P}_{i,h} \widehat{V}_{i,h+1} \right) \right|$$

$$\leq c_5 \sqrt{\frac{H \log^3\left(\frac{KS\sum_i A_i}{\delta}\right)}{K}} 1 + 2c_3 \sqrt{\frac{\log^3\left(\frac{KS\sum_i A_i}{\delta}\right)}{KH}} \left[ \overline{P}_{i,h} \left( \widehat{V}_{i,h+1} \circ \widehat{V}_{i,h+1} \right) - \widehat{V}_{i,h} \circ \widehat{V}_{i,h} \right].$$

(54)

**Step 3: invoking the key recursion to establish the desired bound.** We find it helpful to introduce the following notation (please note the order of the matrix product)

$$\prod_{j:j<h} P_{i,j}^{\widehat{\pi}} := \begin{cases} P_{i,1}^{\widehat{\pi}} \cdots P_{i,h-1}^{\widehat{\pi}}, & \text{if } h > 1, \\ I, & \text{if } h = 1. \end{cases}$$

Armed with this notation, we can invoke the relation (52) recursively and use $\overline{V}_{i,h+1}^{\widehat{\pi}} = V_{i,h+1}^{\widehat{\pi}} = 0$ to obtain

$$\overline{V}_{i,h}^{\widehat{\pi}} - V_{i,h}^{\widehat{\pi}} \overset{(i)}{=} r_{i,h}^{\widehat{\pi}} + P_{i,h}^{\widehat{\pi}} \overline{V}_{i,h+1}^{\widehat{\pi}} + \left( \overline{V}_{i,h}^{\widehat{\pi}} - \left( r_{i,h}^{\widehat{\pi}} + P_{i,h}^{\widehat{\pi}} \overline{V}_{i,h+1}^{\widehat{\pi}} \right) \right) - \left( r_{i,h}^{\widehat{\pi}} + P_{i,h}^{\widehat{\pi}} V_{i,h+1}^{\widehat{\pi}} \right)$$

$$\leq P_{i,h}^{\widehat{\pi}} \left( \overline{V}_{i,h+1}^{\widehat{\pi}} - V_{i,h+1}^{\widehat{\pi}} \right) + \left| \overline{V}_{i,h}^{\widehat{\pi}} - \left( r_{i,h}^{\widehat{\pi}} + P_{i,h}^{\widehat{\pi}} \overline{V}_{i,h+1}^{\widehat{\pi}} \right) \right|$$

(55)

$$\overset{(ii)}{\leq} c_5 \sqrt{\frac{H \log^3\left(\frac{KS\sum_i A_i}{\delta}\right)}{K}} \left( \sum_{h=1}^{H} \prod_{j:j<h} P_{i,j}^{\widehat{\pi}} \right) 1$$

$$+ 2c_3 \sqrt{\frac{\log^3\left(\frac{KS\sum_i A_i}{\delta}\right)}{KH}} \sum_{h=1}^{H} \prod_{j:j<h} P_{i,j}^{\widehat{\pi}} \left[ P_{i,h}^{\widehat{\pi}} \left( \overline{V}_{i,h+1}^{\widehat{\pi}} \circ \overline{V}_{i,h+1}^{\widehat{\pi}} \right) - \overline{V}_{i,h}^{\widehat{\pi}} \circ \overline{V}_{i,h}^{\widehat{\pi}} \right]$$

$$\overset{(iii)}{\leq} c_5 \sqrt{\frac{H \log^3\left(\frac{KS\sum_i A_i}{\delta}\right)}{K}} \left( \sum_{h=1}^{H} \prod_{j:j<h} P_{i,j}^{\widehat{\pi}} \right) 1 = c_5 \sqrt{\frac{H^3 \log^3\left(\frac{KS\sum_i A_i}{\delta}\right)}{K}} 1 \leq \frac{\varepsilon}{3} 1.$$

(56)

Here, (i) uses the Bellman equation; (ii) applies the bound (52) recursively; (iii) holds since for any transition matrices $\{P_{i,h}\}$ and any sequence $\{V_{i,h}\}$ obeying $V_{i,h+1} = 0$, one can use the telescoping sum to obtain

$$\sum_{h=1}^{H} \prod_{j:j<h} P_{i,j} \left[ P_{i,h} \left( V_{i,h+1} \circ V_{i,h+1} \right) - V_{i,h} \circ V_{i,h} \right] = \sum_{h=1}^{H} \prod_{j:j\leq h} P_{i,j} \left( V_{i,h+1} \circ V_{i,h+1} \right) - \sum_{h=1}^{H} \prod_{j:j<h} P_{i,j} \left( V_{i,h} \circ V_{i,h} \right)$$

$$= \prod_{j:j\leq H} P_{i,j} \left( V_{i,h+1} \circ V_{i,h+1} \right) - V_{i,1} \circ V_{i,1}$$

$$= -V_{i,1} \circ V_{i,1} \leq 0,$$

whereas the last inequality in (56) arises from the assumption (27) when $c_k$ is large enough. Similarly, replacing $\widehat{\pi}_i$ with $\widetilde{\pi}_i^\star$ in the above argument and recalling (53) directly lead to

$$V_{i,h}^{\star,\widehat{\pi}_{-i}} - \overline{V}_{i,h}^{\widetilde{\pi}_i^\star \times \widehat{\pi}_{-i}} = V_{i,h}^{\widetilde{\pi}_i^\star \times \widehat{\pi}_{-i}} - \overline{V}_{i,h}^{\widetilde{\pi}_i^\star \times \widehat{\pi}_{-i}} \leq \frac{\varepsilon}{3} 1.$$

(57)

In addition, recalling the definition of $\overline{V}_{i,h}^{\widehat{\pi}}$ (cf. (45c)), $\overline{r}_{i,h}$ and $\overline{P}_{i,h}$ (see (48)), we can deduce that

$$\widehat{V}_{i,h} - \overline{V}_{i,h}^{\widehat{\pi}} = \overline{r}_{i,h} + \overline{P}_{i,h} \widehat{V}_{i,h+1} + \left\{ \widehat{V}_{i,h} - \left( \overline{r}_{i,h} + \overline{P}_{i,h} \widehat{V}_{i,h+1} \right) \right\} - \overline{r}_{i,h} - \overline{P}_{i,h} \overline{V}_{i,h+1}^{\widehat{\pi}}$$

$$\leq \overline{P}_{i,h}\big(\widehat{V}_{i,h+1} - \overline{V}^{\widehat{\pi}}_{i,h+1}\big) + \Big|\widehat{V}_{i,h} - \big(\overline{r}_{i,h} + \overline{P}_{i,h}\widehat{V}_{i,h+1}\big)\Big|,$$

which resembles (55). Thus, repeating the above argument for (56) and applying (54) recursively, we reach

$$\widehat{V}_{i,h} - \overline{V}^{\widehat{\pi}}_{i,h} \leq \frac{\varepsilon}{3}\mathbf{1}. \tag{58}$$

To finish up, combining (56), (57) and (58) with (47), we arrive at

$$V^{\star,\widehat{\pi}-i}_{i,h} - V^{\widehat{\pi}}_{i,h} \leq \big(V^{\star,\widehat{\pi}-i}_{i,h} - \overline{V}^{\widehat{\pi}^{\star}_i \times \widehat{\pi}-i}_{i,h}\big) + \big(\widehat{V}_{i,h} - \overline{V}^{\widehat{\pi}}_{i,h}\big) + \big(\overline{V}^{\widehat{\pi}}_{i,h} - V^{\widehat{\pi}}_{i,h}\big) \leq \varepsilon\mathbf{1}.$$

This establishes the first inequality in (43), while the second inequality in (43) can be validated via the same argument. We have thus completed the proof of Theorem 2.

# D   Proof of Theorem 3

This section is devoted to presenting the proof of Theorem 3. Before embarking on the analysis, let us introduce a convenient auxiliary iterate

$$\pi^-_{k+1} = \arg\min_{\pi\in\Delta(\mathcal{A})}\left\{\langle\pi, L_k\rangle + \frac{1}{\widehat{\eta}_k}F(\pi)\right\}, \tag{59}$$

or equivalently,

$$\pi^-_{k+1}(a) = \frac{\exp\big(-\widehat{\eta}_k L_k(a)\big)}{\sum_{a'\in\mathcal{A}}\exp\big(-\widehat{\eta}_k L_k(a')\big)} \qquad \text{for all } a\in\mathcal{A}, \tag{60}$$

which differs from (34) only in the learning rates being used (namely, $\pi_{k+1}$ uses $\eta_{k+1}$ while $\pi^-_{k+1}$ adopts $\widehat{\eta}_k$).

## D.1   Main steps of the proof

The key steps of the proof lie in justifying the following two claims:

$$R_n \leq \sum_{k=1}^{n}\alpha^n_k\big\langle\pi_k - \pi^-_{k+1}, \ell_k\big\rangle + \frac{\log A}{\eta_{n+1}}; \tag{61}$$

and for all $a\in\mathcal{A}$ and all $k\geq 1$,

$$\pi^-_{k+1}(a) \geq \begin{cases} \big[1 - \widehat{\eta}_k\alpha_k\ell_k(a)\big]\pi_k(a), & \text{if } \widehat{\eta}_k\alpha_k\|\ell_k\|_\infty > \frac{1}{3}, \\ \Big\{1 - \widehat{\eta}_k\alpha_k\big(\ell_k(a) - \mathbb{E}_{\pi_k}[\ell_k]\big) - 2\widehat{\eta}^2_k\alpha^2_k\mathsf{Var}_{\pi_k}\big(\ell_k\big)\Big\}\pi_k(a), & \text{if } \widehat{\eta}_k\alpha_k\|\ell_k\|_\infty \leq \frac{1}{3}, \end{cases} \tag{62}$$

where for any vector $\ell\in\mathbb{R}^A$ we define

$$\mathbb{E}_{\pi_k}[\ell] \coloneqq \sum_{a\in\mathcal{A}}\pi_k(a)\ell(a).$$

In words, the first claim (61) allows us to replace the action that appears best in hindsight (cf. (32)) by the time-varying predictions $\{\pi^-_{k+1}\}$ without incurring much cost, whereas the second claim (62) controls the proximity of $\pi^-_{k+1}$ and $\pi_k$ in each round. Let us assume the validity of these two claims for the moment, and return to prove them shortly.

In view of the upper bound (61), we are in need of controlling $\big\langle\pi_k - \pi^-_{k+1}, \ell_k\big\rangle$. We divide into two cases.

• For any $k$ obeying $\widehat{\eta}_k\alpha_k\|\ell_k\|_\infty > 1/3$, invoke (62) and the non-negativity of $\ell_k$ to reach

$$\big\langle\pi_k - \pi^-_{k+1}, \ell_k\big\rangle \leq \sum_{a\in\mathcal{A}}\widehat{\eta}_k\alpha_k\pi_k(a)\big[\ell_k(a)\big]^2 \leq \sum_{a\in\mathcal{A}}\widehat{\eta}_k\alpha_k\pi_k(a)\big\|\ell_k\big\|^2_\infty = \widehat{\eta}_k\alpha_k\big\|\ell_k\big\|^2_\infty. \tag{63}$$

- In contrast, if $\widehat{\eta}_k \alpha_k \|\ell_k\|_\infty \leq 1/3$, then it follows from (62) that

$$
\begin{aligned}
\left\langle \pi_k - \pi_{k+1}^-, \ell_k \right\rangle &\leq \sum_{a \in \mathcal{A}} \left\{ \widehat{\eta}_k \alpha_k \big(\ell_k(a) - \mathbb{E}_{\pi_k}[\ell_k]\big) + 2\widehat{\eta}_k^2 \alpha_k^2 \mathsf{Var}_{\pi_k}(\ell_k) \right\} \pi_k(a)\ell_k(a) \\
&= \widehat{\eta}_k \alpha_k \sum_{a \in \mathcal{A}} \pi_k(a)\big(\ell_k(a) - \mathbb{E}_{\pi_k}[\ell_k]\big)\mathbb{E}_{\pi_k}[\ell_k] + \widehat{\eta}_k \alpha_k \sum_{a \in \mathcal{A}} \pi_k(a)\big(\ell_k(a) - \mathbb{E}_{\pi_k}[\ell_k]\big)^2 \\
&\quad + 2\widehat{\eta}_k^2 \alpha_k^2 \mathsf{Var}_{\pi_k}(\ell_k) \sum_{a \in \mathcal{A}} \pi_k(a)\ell_k(a) \\
&= \widehat{\eta}_k \alpha_k \sum_{a \in \mathcal{A}} \pi_k(a)\big(\ell_k(a) - \mathbb{E}_{\pi_k}[\ell_k]\big)^2 + 2\widehat{\eta}_k^2 \alpha_k^2 \mathsf{Var}_{\pi_k}(\ell_k) \sum_{a \in \mathcal{A}} \pi_k(a)\ell_k(a) \\
&\leq \widehat{\eta}_k \alpha_k \mathsf{Var}_{\pi_k}(\ell_k) + 2\widehat{\eta}_k^2 \alpha_k^2 \mathsf{Var}_{\pi_k}(\ell_k)\|\ell_k\|_\infty, \tag{64}
\end{aligned}
$$

where we invoke the elementary facts that $\sum_a \pi_k(a)\big(\ell_k(a) - \mathbb{E}_{\pi_k}[\ell_k]\big) = 0$ and $\sum_a \pi_k(a)\ell_k(a) \leq \|\ell_k\|_\infty$.

Putting the above two cases together yields

$$
\begin{aligned}
&\sum_{k=1}^{n} \alpha_k^n \left\langle \pi_k - \pi_{k+1}^-, \ell_k \right\rangle \\
&\leq \sum_{k=1}^{n} \alpha_k^n \widehat{\eta}_k \alpha_k \|\ell_k\|_\infty^2 \mathbb{1}\left(\widehat{\eta}_k \alpha_k \|\ell_k\|_\infty > \frac{1}{3}\right) + \sum_{k=1}^{n} \alpha_k^n \widehat{\eta}_k \alpha_k \mathsf{Var}_{\pi_k}(\ell_k) \mathbb{1}\left(\widehat{\eta}_k \alpha_k \|\ell_k\|_\infty \leq \frac{1}{3}\right) \\
&\quad + 2\sum_{k=1}^{n} \alpha_k^n \widehat{\eta}_k^2 \alpha_k^2 \mathsf{Var}_{\pi_k}(\ell_k)\|\ell_k\|_\infty \mathbb{1}\left(\widehat{\eta}_k \alpha_k \|\ell_k\|_\infty \leq \frac{1}{3}\right) \\
&\leq \frac{5}{3}\sum_{k=1}^{n} \alpha_k^n \widehat{\eta}_k \alpha_k \mathsf{Var}_{\pi_k}(\ell_k) + 3\sum_{k=1}^{n} \alpha_k^n \widehat{\eta}_k^2 \alpha_k^2 \|\ell_k\|_\infty^3 \mathbb{1}\left(\widehat{\eta}_k \alpha_k \|\ell_k\|_\infty > \frac{1}{3}\right), \tag{65}
\end{aligned}
$$

where the last inequality holds true since

$$
\sum_{k=1}^{n} \alpha_k^n \widehat{\eta}_k \alpha_k \|\ell_k\|_\infty^2 \mathbb{1}\left(\widehat{\eta}_k \alpha_k \|\ell_k\|_\infty > \frac{1}{3}\right) \leq 3\sum_{k=1}^{n} \alpha_k^n \widehat{\eta}_k^2 \alpha_k^2 \|\ell_k\|_\infty^3 \mathbb{1}\left(\widehat{\eta}_k \alpha_k \|\ell_k\|_\infty > \frac{1}{3}\right),
$$

$$
\sum_{k=1}^{n} \alpha_k^n \widehat{\eta}_k^2 \alpha_k^2 \|\ell_k\|_\infty \mathsf{Var}_{\pi_k}(\ell_k) \mathbb{1}\left(\widehat{\eta}_k \alpha_k \|\ell_k\|_\infty \leq \frac{1}{3}\right) \leq \frac{1}{3}\sum_{k=1}^{n} \alpha_k^n \widehat{\eta}_k \alpha_k \mathsf{Var}_{\pi_k}(\ell_k).
$$

Substituting (65) into (61), we can readily arrive at

$$
R_n \leq \frac{5}{3}\sum_{k=1}^{n} \alpha_k^n \widehat{\eta}_k \alpha_k \mathsf{Var}_{\pi_k}(\ell_k) + \frac{\log A}{\eta_{n+1}} + 3\sum_{k=1}^{n} \alpha_k^n \widehat{\eta}_k^2 \alpha_k^2 \|\ell_k\|_\infty^3 \mathbb{1}\left(\widehat{\eta}_k \alpha_k \|\ell_k\|_\infty > \frac{1}{3}\right).
$$

It thus remains to establish the claims (61) and (62), which we shall accomplish next.

### D.2  Proof of claim (61)

We claim that it suffices to prove that

$$
\begin{aligned}
&\alpha_1^n \langle \pi_2^-, \ell_1 \rangle + \frac{\alpha_1^n}{\eta_2 \alpha_1} F(\pi_2) + \sum_{k=2}^{n} \left\{ \alpha_k^n \langle \pi_{k+1}^-, \ell_k \rangle + \left[\frac{\alpha_k^n}{\eta_{k+1}\alpha_k} - \frac{\alpha_{k-1}^n}{\eta_k \alpha_{k-1}}\right] F(\pi_{k+1}) \right\} \\
&\qquad\qquad \leq \min_{\pi \in \Delta(\mathcal{A})} \left\{ \left\langle \pi, \sum_{k=1}^{n} \alpha_k^n \ell_k \right\rangle + \frac{1}{\eta_{n+1}} F(\pi) \right\}. \tag{66}
\end{aligned}
$$

In fact, suppose that this inequality (66) is valid, then one can easily obtain

$$
\alpha_1^n \langle \pi_2^-, \ell_1 \rangle + \frac{\alpha_1^n}{\eta_2 \alpha_1} F(\pi_2) + \sum_{k=2}^{n} \left\{ \alpha_k^n \langle \pi_{k+1}^-, \ell_k \rangle + \left[\frac{\alpha_k^n}{\eta_{k+1}\alpha_k} - \frac{\alpha_{k-1}^n}{\eta_k \alpha_{k-1}}\right] F(\pi_{k+1}) \right\}
$$

$$\leq \min_{\pi \in \Delta(\mathcal{A})} \left\{ \left\langle \pi, \sum_{k=1}^{n} \alpha_k^n \ell_k \right\rangle + \frac{1}{\eta_{n+1}} F(\pi) \right\} \leq \min_{\pi \in \{e_a \mid a \in \mathcal{A}\}} \left\{ \left\langle \pi, \sum_{k=1}^{n} \alpha_k^n \ell_k \right\rangle + \frac{1}{\eta_{n+1}} F(\pi) \right\}$$

$$= \min_{\pi \in \{e_a \mid a \in \mathcal{A}\}} \left\langle \pi, \sum_{k=1}^{n} \alpha_k^n \ell_k \right\rangle = \min_{a \in \mathcal{A}} \sum_{k=1}^{n} \alpha_k^n \ell_k(a)$$

with $e_a$ the $a$-th standard basis vector in $\mathbb{R}^A$, where the last line holds true since the negative entropy obeys $F(e_a) = 0$ for any $a \in \mathcal{A}$. In turn, this implies that

$$R_n = \sum_{k=1}^{n} \alpha_k^n \langle \pi_k, \ell_k \rangle - \min_{a \in \mathcal{A}} \sum_{k=1}^{n} \alpha_k^n \ell_k(a)$$

$$\leq \sum_{k=1}^{n} \alpha_k^n \langle \pi_k - \pi_{k+1}^-, \ell_k \rangle - \sum_{k=2}^{n} \left[ \frac{\alpha_k^n}{\eta_{k+1} \alpha_k} - \frac{\alpha_{k-1}^n}{\eta_k \alpha_{k-1}} \right] F(\pi_{k+1}) + \frac{\alpha_1^n}{\eta_2 \alpha_1} \log A, \qquad (67)$$

where the last inequality invokes the elementary fact $-F(\pi) \leq \log A$ for any $\pi \in \Delta(\mathcal{A})$. Additionally, under the assumptions that $\eta_{k+1}(1 - \alpha_k) \leq \eta_k$ ($k \geq 1$), we can use the definition (5) to obtain

$$\frac{\alpha_k^n}{\eta_{k+1} \alpha_k} = \frac{\prod_{j=k+1}^{n}(1 - \alpha_j)}{\eta_{k+1}} \geq \frac{\prod_{j=k}^{n}(1 - \alpha_j)}{\eta_k} = \frac{\alpha_{k-1}^n}{\eta_k \alpha_{k-1}},$$

for any $k \geq 2$, which together with the basic fact $0 \leq -F(\pi) \leq \log A$ yields

$$-\sum_{k=2}^{n} \left[ \frac{\alpha_k^n}{\eta_{k+1} \alpha_k} - \frac{\alpha_{k-1}^n}{\eta_k \alpha_{k-1}} \right] F(\pi_{k+1}) + \frac{\alpha_1^n}{\eta_2 \alpha_1} \log A \leq \sum_{k=2}^{n} \left[ \frac{\alpha_k^n}{\eta_{k+1} \alpha_k} - \frac{\alpha_{k-1}^n}{\eta_k \alpha_{k-1}} \right] \log A + \frac{\alpha_1^n}{\eta_2 \alpha_1} \log A$$

$$= \frac{\alpha_n^n}{\eta_{n+1} \alpha_n} \log A = \frac{\log A}{\eta_{n+1}}. \qquad (68)$$

Substitution into (67) leads to

$$R_n \leq \sum_{k=1}^{n} \alpha_k^n \langle \pi_k - \pi_{k+1}^-, \ell_k \rangle + \frac{\log A}{\eta_{n+1}} \qquad (69)$$

as advertised. As a consequence, everything boils down to establishing (66).

Towards this end, we would like to proceed with an induction argument, with the induction hypothesis w.r.t. $n$ given by (66). Firstly, the base case with $n = 1$ simplifies to

$$\alpha_1^1 \langle \pi_2^-, \ell_1 \rangle + \frac{1}{\eta_2} F(\pi_2) \leq \min_{\pi \in \Delta(\mathcal{A})} \left\{ \langle \pi, \alpha_1^1 \ell_1 \rangle + \frac{1}{\eta_2} F(\pi) \right\}$$

given that $\alpha_1 = \alpha_1^1$; this inequality clearly holds since, according to (33) and (59),

$$\pi_2^- = \pi_2 = \arg \min_{\pi \in \Delta(\mathcal{A})} \left\{ \langle \pi, L_1 \rangle + \frac{1}{\eta_2} F(\pi) \right\} = \arg \min_{\pi \in \Delta(\mathcal{A})} \left\{ \langle \pi, \alpha_1 \ell_1 \rangle + \frac{1}{\eta_2} F(\pi) \right\}.$$

Secondly, suppose that (66) holds w.r.t. $n$, and we intend to justify it w.r.t. $n + 1$. To do so, we observe that

$$\alpha_1^{n+1} \langle \pi_2^-, \ell_1 \rangle + \frac{\alpha_1^{n+1}}{\eta_2 \alpha_1} F(\pi_2) + \sum_{k=2}^{n} \left\{ \alpha_k^{n+1} \langle \pi_{k+1}^-, \ell_k \rangle + \left( \frac{\alpha_k^{n+1}}{\eta_{k+1} \alpha_k} - \frac{\alpha_{k-1}^{n+1}}{\eta_k \alpha_{k-1}} \right) F(\pi_{k+1}) \right\} + \alpha_{n+1} \langle \pi_{n+2}^-, \ell_{n+1} \rangle$$

$$\overset{\text{(i)}}{=} (1 - \alpha_{n+1}) \left\{ \alpha_1^n \langle \pi_2^-, \ell_1 \rangle + \frac{\alpha_1^n}{\eta_2 \alpha_1} F(\pi_2) + \sum_{k=2}^{n} \left\{ \alpha_k^n \langle \pi_{k+1}^-, \ell_k \rangle + \left( \frac{\alpha_k^n}{\eta_{k+1} \alpha_k} - \frac{\alpha_{k-1}^n}{\eta_k \alpha_{k-1}} \right) F(\pi_{k+1}) \right\} \right\}$$

$$\qquad + \alpha_{n+1} \langle \pi_{n+2}^-, \ell_{n+1} \rangle$$

$$\overset{\text{(ii)}}{\leq} (1 - \alpha_{n+1}) \left\{ \left\langle \pi_{n+2}^-, \sum_{k=1}^{n} \alpha_k^n \ell_k \right\rangle + \frac{1}{\eta_{n+1}} F(\pi_{n+2}^-) \right\} + \alpha_{n+1} \langle \pi_{n+2}^-, \ell_{n+1} \rangle$$

$$\overset{\text{(iii)}}{=} \left\langle \pi_{n+2}^-, \sum_{k=1}^{n+1} \alpha_k^{n+1} \ell_k \right\rangle + \frac{1 - \alpha_{n+1}}{\eta_{n+1}} F(\pi_{n+2}^-) = \min_{\pi \in \Delta(\mathcal{A})} \left\{ \left\langle \pi, \sum_{k=1}^{n+1} \alpha_k^{n+1} \ell_k \right\rangle + \frac{1}{\widehat{\eta}_{n+1}} F(\pi) \right\}.$$

$$(70)$$

Here, (i) and (iii) invoke the fact $\alpha_k^{n+1} = (1 - \alpha_{n+1})\alpha_k^n$ and $\alpha_{n+1}^{n+1} = \alpha_{n+1}$ (according to (5)), (ii) relies on the induction hypothesis (66) w.r.t. $n$. To finish up, invoke (70) and the definition (5) to arrive at

$$\alpha_1^{n+1}\langle \pi_2^-, \ell_1 \rangle + \frac{\alpha_1^{n+1}}{\eta_2 \alpha_1} F(\pi_2) + \sum_{k=2}^{n+1} \left\{ \alpha_k^{n+1}\langle \pi_{k+1}^-, \ell_k \rangle + \left[ \frac{\alpha_k^{n+1}}{\eta_{k+1}\alpha_k} - \frac{\alpha_{k-1}^{n+1}}{\eta_k \alpha_{k-1}} \right] F(\pi_{k+1}) \right\}$$

$$= \left\{ \alpha_1^{n+1}\langle \pi_2^-, \ell_1 \rangle + \frac{\alpha_1^{n+1}}{\eta_2 \alpha_1} F(\pi_2) + \sum_{k=2}^{n} \left\{ \alpha_k^{n+1}\langle \pi_{k+1}^-, \ell_k \rangle + \left[ \frac{\alpha_k^{n+1}}{\eta_{k+1}\alpha_k} - \frac{\alpha_{k-1}^{n+1}}{\eta_k \alpha_{k-1}} \right] F(\pi_{k+1}) \right\} + \alpha_{n+1}\langle \pi_{n+2}^-, \ell_{n+1} \rangle \right\}$$

$$+ \left[ \frac{1}{\eta_{n+2}} - \frac{1 - \alpha_{n+1}}{\eta_{n+1}} \right] F(\pi_{n+2})$$

$$\leq \left\{ \left\langle \pi_{n+2}, \sum_{k=1}^{n+1} \alpha_k^{n+1}\ell_k \right\rangle + \frac{1 - \alpha_{n+1}}{\eta_{n+1}} F(\pi_{n+2}) \right\} + \left[ \frac{1}{\eta_{n+2}} - \frac{1 - \alpha_{n+1}}{\eta_{n+1}} \right] F(\pi_{n+2})$$

$$= \left\langle \pi_{n+2}, \sum_{k=1}^{n+1} \alpha_k^{n+1}\ell_k \right\rangle + \frac{1}{\eta_{n+2}} F(\pi_{n+2}) = \min_{\pi \in \Delta(\mathcal{A})} \left\{ \left\langle \pi, \sum_{k=1}^{n+1} \alpha_k^{n+1}\ell_k \right\rangle + \frac{1}{\eta_{n+2}} F(\pi) \right\},$$

where the inequality above makes use of (70), and the last identity comes from (33). This justifies the induction hypothesis w.r.t. $n + 1$. Applying the induction argument in turn establishes (66) for all $n$, thereby concluding the proof.

### D.3 Proof of claim (62)

We first make the observation that

$$\sum_a \exp\left( -\widehat{\eta}_k L_k(a) \right) = \sum_a \exp\left( -\eta_k L_{k-1}(a) \right) \exp\left( -\widehat{\eta}_k \alpha_k \ell_k(a) \right)$$

$$= \sum_a \left\{ \pi_k(a) \sum_{a'} \exp\left( -\eta_k L_{k-1}(a') \right) \right\} \exp\left( -\widehat{\eta}_k \alpha_k \ell_k(a) \right)$$

$$= \sum_{a'} \exp\left( -\eta_k L_{k-1}(a') \right) \sum_a \left\{ \pi_k(a) \exp\left( -\widehat{\eta}_k \alpha_k \ell_k(a) \right) \right\},$$

where the second equality follows from (34). This in turn allows us to demonstrate that

$$\pi_{k+1}^-(a) = \frac{\exp\left( -\widehat{\eta}_k L_k(a) \right)}{\sum_{a'} \exp\left( -\widehat{\eta}_k L_k(a') \right)} = \frac{\exp\left( -\eta_k L_{k-1}(a) \right)}{\sum_{a'} \exp\left( -\eta_k L_{k-1}(a') \right)} \cdot \frac{\exp\left( -\widehat{\eta}_k \alpha_k \ell_k(a) \right)}{\sum_{a'} \pi_k(a') \exp\left( -\widehat{\eta}_k \alpha_k \ell_k(a') \right)}$$

$$= \pi_k(a) \frac{\exp\left( -\widehat{\eta}_k \alpha_k \ell_k(a) \right)}{\sum_{a'} \pi_k(a') \exp\left( -\widehat{\eta}_k \alpha_k \ell_k(a') \right)} \geq \left[ 1 - \widehat{\eta}_k \alpha_k \ell_k(a) \right] \pi_k(a),$$

where the last inequality holds since $\exp(-x) \geq 1 - x$ and $\sum_a \pi_k(a) \exp\left( -\widehat{\eta}_k \alpha_k \ell_k(a) \right) \leq \sum_a \pi_k(a) = 1$.

Next, suppose that $\widehat{\eta}_k \alpha_k \|\ell_k\|_\infty \leq 1/3$. In this case, it is self-evident that $\widehat{\eta}_k \alpha_k |\ell_k(a) - \mathbb{E}_{\pi_k}[\ell_k]| \leq 2/3$ for all $a \in \mathcal{A}$. Recalling that $\mathbb{E}_{\pi_k}[\ell_k] = \sum_a \pi_k(a)\ell_k(a)$, one can derive

$$\pi_{k+1}^-(a) = \pi_k(a) \frac{\exp\left( -\widehat{\eta}_k \alpha_k \ell_k(a) \right)}{\sum_{a'} \pi_k(a') \exp\left( -\widehat{\eta}_k \alpha_k \ell_k(a') \right)} = \frac{\exp\left( -\widehat{\eta}_k \alpha_k \left( \ell_k(a) - \mathbb{E}_{\pi_k}[\ell_k] \right) \right)}{\sum_{a'} \pi_k(a') \exp\left( -\widehat{\eta}_k \alpha_k \left( \ell_k(a') - \mathbb{E}_{\pi_k}[\ell_k] \right) \right)} \pi_k(a)$$

$$\geq \frac{1 - \widehat{\eta}_k \alpha_k \left( \ell_k(a) - \mathbb{E}_{\pi_k}[\ell_k] \right)}{\sum_{a'} \pi_k(a') \exp\left( -\widehat{\eta}_k \alpha_k \left( \ell_k(a') - \mathbb{E}_{\pi_k}[\ell_k] \right) \right)} \pi_k(a)$$

$$\geq \frac{1 - \widehat{\eta}_k \alpha_k \left( \ell_k(a) - \mathbb{E}_{\pi_k}[\ell_k] \right)}{1 + \widehat{\eta}_k^2 \alpha_k^2 \mathsf{Var}_{\pi_k}(\ell_k)} \pi_k(a); \tag{71}$$

here, the first inequality arises since $\exp(-x) \geq 1 - x$, while the second inequality can be shown via the elementary inequality $\exp(-x) \leq 1 - x + x^2$ for any $x \geq -1.5$ and therefore

$$\sum_a \pi_k(a) \exp\left( -\widehat{\eta}_k \alpha_k \left( \ell_k(a) - \mathbb{E}_{\pi_k}[\ell_k] \right) \right)$$

$$\leq \sum_a \pi_k(a) \left\{ 1 - \widehat{\eta}_k \alpha_k \big( \ell_k(a) - \mathbb{E}_{\pi_k}[\ell_k] \big) + \widehat{\eta}_k^2 \alpha_k^2 \big( \ell_k(a) - \mathbb{E}_{\pi_k}[\ell_k] \big)^2 \right\}$$

$$= \sum_a \pi_k(a) \left\{ 1 + \widehat{\eta}_k^2 \alpha_k^2 \big( \ell_k(a) - \mathbb{E}_{\pi_k}[\ell_k] \big)^2 \right\}$$

$$= 1 + \widehat{\eta}_k^2 \alpha_k^2 \mathsf{Var}_{\pi_k}(\ell_k).$$

Applying the elementary inequality $\frac{1-a}{1+b} \geq (1-a)(1-b) = 1 - a - b + ab \geq 1 - a - 2b$ for any $a \in [-1, 1]$ and $b > 0$, we can continue to lower bound (71) as follows

$$(71) \geq \left\{ 1 - \widehat{\eta}_k \alpha_k \big( \ell_k(a) - \mathbb{E}_{\pi_k}[\ell_k] \big) - 2 \widehat{\eta}_k^2 \alpha_k^2 \mathsf{Var}_{\pi_k}(\ell_k) \right\} \pi_k(a),$$

thereby completing the proof.

# E  Proofs of auxiliary lemmas and details

## E.1  Proof of Lemma 3

This section aims to prove Lemma 3, which establishes the inequality $\widehat{V}_{i,h} \geq \overline{V}_{i,h}^{\star, \widehat{\pi}_{-i}}$. In what follows, we shall proceed with an induction argument. The base case with step $H + 1$ is trivially true, given that

$$\widehat{V}_{i,H+1} = \overline{V}_{i,H+1}^{\star, \widehat{\pi}_{-i}} = 0$$

holds for any joint policy. Next, let us assume that the claim (46) is valid for step $h + 1$, namely,

$$\widehat{V}_{i,h+1} \geq \overline{V}_{i,h+1}^{\star, \widehat{\pi}_{-i}}, \tag{72}$$

and attempt to justify the validity of this result when $h + 1$ is replaced with $h$.

This step is mainly accomplished by applying our refined theory (cf. Theorem 3) for FTRL (see (19)). More precisely, we claim that

$$\max_{a_i} Q_{i,h}^K(s, a_i) \leq \sum_{k=1}^K \alpha_k^K \left\langle \pi_{i,h}^k(s), \, q_{i,h}^k(s, \cdot) \right\rangle$$

$$+ 10 \sqrt{\frac{c_\alpha \log^3(KA_i)}{KH} \sum_{k=1}^K \alpha_k^K \mathsf{Var}_{\pi_{i,h}^k(s)} \big( q_{i,h}^k(s, \cdot) \big)} + 2 \sqrt{\frac{c_\alpha H \log^3(KA_i)}{K}} \tag{73}$$

for any $s \in \mathcal{S}$, whose proof is deferred to Appendix E.1.1. Recall the construction (20a) of $\widehat{V}_{i,h}$. If $\widehat{V}_{i,h} = H - h + 1$, then the claimed result $\widehat{V}_{i,h} \geq \overline{V}_{i,h}^{\star, \widehat{\pi}_{-i}}$ holds trivially. It thus suffices to focus on the case where

$$\widehat{V}_{i,h}(s) = \sum_{k=1}^K \alpha_k^K \left\langle \pi_{i,h}^k(s), \, q_{i,h}^k(s, \cdot) \right\rangle + \beta_{i,h}(s). \tag{74}$$

In this case, recalling the definition of $\overline{V}_{i,h}^{\star, \widehat{\pi}_{-i}}(s)$ in (45b) gives

$$\overline{V}_{i,h}^{\star, \widehat{\pi}_{-i}}(s) = \max_{a_i} \sum_{k=1}^K \alpha_k^K \left[ r_{i,h}^k(s, a_i) + \left\langle P_{i,h}^k(\cdot \mid s, a_i), \overline{V}_{i,h+1}^{\star, \widehat{\pi}_{-i}} \right\rangle \right]$$

$$\leq \max_{a_i} \sum_{k=1}^K \alpha_k^K \left[ r_{i,h}^k(s, a_i) + \left\langle P_{i,h}^k(\cdot \mid s, a_i), \widehat{V}_{i,h+1} \right\rangle \right] = \max_{a_i} Q_{i,h}^K(s, a_i)$$

$$\leq \sum_{k=1}^K \alpha_k^K \left\langle \pi_{i,h}^k(s), \, q_{i,h}^k(s, \cdot) \right\rangle + 10 \sqrt{\frac{c_\alpha \log^3(KA_i)}{KH} \sum_{k=1}^K \alpha_k^K \mathsf{Var}_{\pi_{i,h}^k(s)} \big( q_{i,h}^k(s, \cdot) \big)} + 2 \sqrt{\frac{c_\alpha H \log^3(KA_i)}{K}}$$

$$\leq \sum_{k=1}^{K} \alpha_k^K \left\langle \pi_{i,h}^k(s), q_{i,h}^k(s,\cdot) \right\rangle + \beta_{i,h}(s) = \widehat{V}_{i,h}(s)$$

simultaneously for all $(s, h) \in \mathcal{S} \times [H]$. Here, the second line follows from the induction hypothesis (72) and the definition (17) of $Q_{i,h}^K$, the third line invokes the claim (73), whereas the last line comes from our choice (24) of $\beta_{i,h}$ (provided $c_b$ is large enough) and (74). This concludes the proof, as long as (73) can be justified.

### E.1.1 Proof of claim (73)

Consider any state $s \in \mathcal{S}$. By virtue of the identity $Q_{i,h}^k = \sum_{j=1}^{k} \alpha_j^k q_{i,h}^j$ (see (17)), the policy update rule (18) (or (19)) for $\pi_{i,h}^k(s)$ can essentially be viewed as the FTRL algorithm applied to the loss vectors

$$\ell_k = -q_{i,h}^k(s,\cdot), \qquad k \geq 1.$$

Moreover, recalling the definition (23) of $\eta_{k+1}$ and the definition (22) of $\alpha_k$ (with $c_\alpha \geq 24$), we have

$$\left( \frac{\eta_k}{\eta_{k+1}} \right)^2 = \frac{\alpha_k}{\alpha_{k-1}} = \frac{k - 2 + c_\alpha \log K}{k - 1 + c_\alpha \log K} \geq \frac{k-1}{k - 1 + c_\alpha \log K} = 1 - \alpha_k > (1 - \alpha_k)^2. \tag{75}$$

This property (75) permits us to invoke Theorem 3 to obtain

$$\max_{a_i \in \mathcal{A}_i} Q_{i,h}^K(s, a_i) - \sum_{k=1}^{K} \alpha_k^K \left\langle \pi_{i,h}^k(s), q_{i,h}^k(s,\cdot) \right\rangle = \max_{a_i \in \mathcal{A}_i} \left\{ \sum_{k=1}^{K} \alpha_k^K \left\langle \pi_{i,h}^k(s), \ell_k \right\rangle - \sum_{k=1}^{K} \alpha_k^K \ell_k(a_i) \right\}$$

$$\leq \frac{5}{3} \sum_{k=2}^{K} \alpha_k^K \frac{\eta_k \alpha_k}{1 - \alpha_k} \mathsf{Var}_{\pi_{i,h}^k(s)}\left( q_{i,h}^k(s,\cdot) \right) + \frac{\log A_i}{\eta_{K+1}} + \xi_{i,h}$$

$$\overset{(i)}{\leq} \frac{5}{3} \sum_{k=2}^{K/2} \frac{(2c_\alpha)^{1.5} \log^2 K}{\sqrt{kH}} \alpha_k^K \mathsf{Var}_{\pi_{i,h}^k(s)}\left( q_{i,h}^k(s,\cdot) \right)$$

$$+ \frac{20}{3} \sum_{k=K/2+1}^{K} \alpha_k^K \sqrt{\frac{c_\alpha \log^2 K}{KH}} \mathsf{Var}_{\pi_{i,h}^k(s)}\left( q_{i,h}^k(s,\cdot) \right) + \frac{\log A_i}{\eta_{K+1}} + \xi_{i,h}, \tag{76}$$

where $\xi_{i,h}$ is defined as

$$\xi_{i,h} := \frac{5}{3} \alpha_1^K \eta_2 \left\| q_{i,h}^1 \right\|_\infty^2 + \left\{ 3 \sum_{k=2}^{K} \alpha_k^K \frac{\eta_k^2 \alpha_k^2}{(1 - \alpha_k)^2} \left\| q_{i,h}^k \right\|_\infty^3 \mathbb{1}\left( \frac{\eta_k \alpha_k}{1 - \alpha_k} \left\| q_{i,h}^k \right\|_\infty > \frac{1}{3} \right) \right\} + 3\alpha_1^K \eta_2^2 \left\| q_{i,h}^1 \right\|_\infty^3. \tag{77}$$

Here, to see why (i) holds, we make use of the facts that

$$1 - \alpha_k = 1 - \frac{c_\alpha \log K}{k - 1 + c_\alpha \log K} \geq \begin{cases} 1 - \frac{c_\alpha \log K}{1 + c_\alpha \log K} = \frac{1}{1 + c_\alpha \log K} \geq \frac{1}{2 c_\alpha \log K}, & \text{if } k \geq 2, \\ 1 - \frac{c_\alpha \log K}{K/2 + c_\alpha \log K} = \frac{K}{K + 2 c_\alpha \log K} \geq \frac{1}{2}, & \text{if } k \geq K/2 + 1, \end{cases} \tag{78a}$$

$$\eta_k \alpha_k = \sqrt{\frac{\log K}{\alpha_{k-1} H}} \cdot \alpha_k \leq \sqrt{\frac{\log K}{\alpha_k H}} \cdot \alpha_k = \sqrt{\frac{\alpha_k \log K}{H}} \leq \sqrt{\frac{2 c_\alpha \log^2 K}{kH}}, \tag{78b}$$

where the first line makes use of (38) for large enough $c_k$, and the second line relies on (39a) in Lemma 1.

To proceed, let us control the terms in (76) separately.

- We start with the first term in (76). The elementary bound $\left\| q_{i,h}^k \right\|_\infty \leq H$ in Lemma 2 taken together with (39b) in Lemma 1 helps us derive

$$\sum_{k=2}^{K/2} \frac{\alpha_k^K \log^2 K}{\sqrt{kH}} \mathsf{Var}_{\pi_{i,h}^k(s)}\left( q_{i,h}^k(s,\cdot) \right) \leq \sum_{k=2}^{K/2} \frac{\log^2 K}{K^6 \sqrt{kH}} \mathsf{Var}_{\pi_{i,h}^k(s)}\left( q_{i,h}^k(s,\cdot) \right)$$

$$\leq \sum_{k=2}^{K/2} \frac{\log^2 K}{K^6 \sqrt{kH}} \big\| q_{i,h}^k(s,\cdot) \big\|_\infty^2 \leq \frac{H^{3/2} \log^2 K}{K^6} \sum_{k=2}^{K/2} \frac{1}{\sqrt{k}}$$

$$\leq \frac{2H^{3/2} \log^2 K}{K^6} \cdot \sqrt{K/2} \leq \frac{2H^{3/2} \log^2 K}{K^5}. \tag{79}$$

- Turning to the third term in (76), we recall the definition of $\eta_{K+1}$ (cf. (23)) to obtain

$$\frac{\log A_i}{\eta_{K+1}} = \log A_i \sqrt{\frac{\alpha_K H}{\log K}} \leq \sqrt{\frac{2c_\alpha H \log^2 A_i}{K}}, \tag{80}$$

where the inequality comes from Lemma 1.

- Finally, we move on to the last term in (76). For any $k \geq 2$, combine Lemma 2 with (78) to obtain

$$\frac{\eta_k \alpha_k}{1 - \alpha_k} \big\| q_{i,h}^k \big\|_\infty \leq \frac{\sqrt{\frac{2c_\alpha \log^2 K}{kH}}}{\frac{1}{2c_\alpha \log K}} \cdot H = \sqrt{\frac{8c_\alpha^3 H \log^4 K}{k}}. \tag{81}$$

Clearly, the right-hand side of (81) is upper bounded by $1/3$ for all $k$ obeying $k \geq c_9 H \log^4 \frac{K}{\delta}$ for some large enough constant $c_9 > 0$ (see also (38)). Consequently, one can derive

$$\xi_{i,h} = \frac{5}{3} \alpha_1^K \eta_2 \big\| q_{i,h}^1 \big\|_\infty^2 + \left\{ 3 \sum_{k=2}^K \alpha_k^K \frac{\eta_k^2 \alpha_k^2}{(1-\alpha_k)^2} \big\| q_{i,h}^k \big\|_\infty^3 \mathbb{1}\left( \frac{\eta_k \alpha_k}{1-\alpha_k} \big\| q_{i,h}^k \big\|_\infty > \frac{1}{3} \right) \right\} + 3\alpha_1^K \eta_2^2 \big\| q_{i,h}^1 \big\|_\infty^3$$

$$\leq \frac{5}{3K^6} \sqrt{\frac{\log K}{H}} \big\| q_{i,h}^1 \big\|_\infty^2 + \frac{(2c_\alpha \log K)^2}{K^6} \left\{ 3 \sum_{k=2}^{c_9 H \log^4 \frac{K}{\delta}} \eta_k^2 \alpha_k^2 \big\| q_{i,h}^k \big\|_\infty^3 \right\} + \frac{3}{K^6} \frac{\log K}{H} \big\| q_{i,h}^1 \big\|_\infty^3$$

$$\leq \frac{24c_\alpha^3 \log^4 K}{K^6 H} \left\{ \sum_{k=1}^K \frac{1}{k} H^3 \right\}$$

$$\leq \frac{24c_\alpha^3 H^2 \log^5 K}{K^6} \leq \frac{1}{K^4}, \tag{82}$$

where the second line comes from (78) and the fact that $K/2 > c_9 H \log^4 \frac{K}{\delta}$ (as a consequence of (38)), and the third line holds due to Lemma 2.

Putting the preceding bounds together and substituting them into (76), we arrive at

$$\max_{a_i} Q_{i,h}^K(s, a_i) - \sum_{k=1}^K \alpha_k^K \left\langle \pi_{i,h}^k(s), q_{i,h}^k(s,\cdot) \right\rangle$$

$$\leq \frac{5(2c_\alpha)^{1.5}}{3} \cdot \frac{2H^{3/2} \log^2 K}{K^5} + \frac{20}{3} \sqrt{\frac{c_\alpha \log^2 K}{KH}} \sum_{k=K/2+1}^K \alpha_k^K \mathsf{Var}_{\pi_{i,h}^k(s)} \left( q_{i,h}^k(s,\cdot) \right) + \sqrt{\frac{2c_\alpha H \log^2 A_i}{K}} + \frac{1}{K^4}$$

$$\leq 10 \sqrt{\frac{c_\alpha \log^3(KA_i)}{KH}} \sum_{k=1}^K \alpha_k^K \mathsf{Var}_{\pi_{i,h}^k(s)} \left( q_{i,h}^k(s,\cdot) \right) + 2\sqrt{\frac{c_\alpha H \log^3(KA_i)}{K}}, \tag{83}$$

where the last line is valid under Condition (38). This completes the proof of Claim (73).

### E.2    Proof of Lemma 4

In this section, we present the proof of Lemma 4. To begin with, we introduce the auxiliary quantities

$$\widetilde{q}_{i,h}^k(s, a_i) := r_{i,h}^k(s, a_i) + P_{i,h}^k(\cdot \,|\, s, a_i) \overline{V}_{i,h+1}^{\widehat{\pi}}, \qquad \forall (s, a_i) \in \mathcal{S} \times \mathcal{A}_i.$$

It is also helpful to introduce an auxiliary random action $a_{k,s} \in \mathcal{A}_i$ generated in a way that

$$a_{k,s} \sim \pi_{i,h}^k(s),$$

which is independent from $\widetilde{q}_{i,h}^k$ conditional on $\pi_{i,h}^k$. This allows us to define another set of random variables

$$\widehat{q}_{i,h}^k(s) \coloneqq \widetilde{q}_{i,h}^k(s, a_{k,s}), \qquad \forall s \in \mathcal{S}, \tag{84}$$

which plays a central role in our analysis. It is readily seen from the facts $\overline{V}_{i,h+1}(s) \le H - h$ (cf. (20a)) and $r_{i,h}^k(s, a_i) \in [0, 1]$ that

$$0 \le \widehat{q}_{i,h}^k(s), \widetilde{q}_{i,h}^k(s, a_i) \le H - h + 1, \qquad \forall (s, a_i, h, k) \in \mathcal{S} \times \mathcal{A}_i \times [H] \times [K]. \tag{85}$$

Letting $e(i) \in \mathbb{R}^{A_i}$ denote the $i$-th standard basis vector, we learn from the law of total variance that

$$
\begin{aligned}
\mathsf{Var}_{h,k-1}\Big(\widehat{q}_{i,h}^k(s)\Big) &= \mathsf{Var}_{h,k-1}\Big(\big\langle e(a_{k,s}), \widetilde{q}_{i,h}^k(s, \cdot)\big\rangle\Big) \\
&\ge \mathsf{Var}_{h,k-1}\Big(\mathbb{E}_{h,k-1}\big[\big\langle e(a_{k,s}), \widetilde{q}_{i,h}^k(s, \cdot)\big\rangle \mid \widetilde{q}_{i,h}^k\big]\Big) \\
&= \mathsf{Var}_{h,k-1}\Big(\big\langle \pi_{i,h}^k(s), \widetilde{q}_{i,h}^k(s, \cdot)\big\rangle\Big).
\end{aligned}
\tag{86}
$$

With these preparations in place, we are ready to embark on the proof.

### E.2.1 Proof of inequalities (49a) and (49b)

Recall the definition of $\overline{V}_{i,h}^{\widehat{\pi}}(s)$ in (45c) that

$$\overline{V}_{i,h}^{\widehat{\pi}}(s) = \sum_{k=1}^K \alpha_k^K \mathop{\mathbb{E}}_{a_i \sim \pi_{i,h}^k(s)} \Big[ r_{i,h}^k(s, a_i) + P_{i,h}^k(\cdot \mid s, a_i)\overline{V}_{i,h+1}^{\widehat{\pi}} \Big] = \sum_{k=1}^K \alpha_k^K \big\langle \pi_{i,h}^k(s), \widetilde{q}_{i,h}^k(s, \cdot)\big\rangle. \tag{87}$$

It is first observed that

$$
\begin{aligned}
\sum_{k=1}^K \mathbb{E}_{h,k-1}\Big[\alpha_k^K \big\langle \pi_{i,h}^k(s), \widetilde{q}_{i,h}^k(s, \cdot)\big\rangle\Big] &= \sum_{k=1}^K \alpha_k^K \mathop{\mathbb{E}}_{\boldsymbol{a} \sim \pi_h^k(s)} \Big[ r_{i,h}(s, \boldsymbol{a}) + \big\langle P_{i,h}(\cdot \mid s, \boldsymbol{a}), \overline{V}_{i,h+1}^{\widehat{\pi}}\big\rangle \mid \overline{V}_{i,h+1}^{\widehat{\pi}}, \pi_{i,h}^k \Big] \\
&= r_{i,h}^{\widehat{\pi}}(s) + \big\langle P_{i,h}^{\widehat{\pi}}(s, \cdot), \overline{V}_{i,h+1}^{\widehat{\pi}}\big\rangle,
\end{aligned}
\tag{88}
$$

where the second identity arises from the definitions (48) of $r_{i,h}^{\widehat{\pi}}$ and $P_{i,h}^{\widehat{\pi}}$. It is also seen that

$$R_1 \coloneqq \max_k \Big|\alpha_k^K \big\langle \pi_{i,h}^k(s), \widetilde{q}_{i,h}^k(s, \cdot)\big\rangle\Big| \le \Big\{ \max_k \alpha_k^K \Big\} \Big\{ \max_k \big\|\pi_{i,h}^k(s)\big\|_1 \big\|\widetilde{q}_{i,h}^k\big\|_\infty \Big\} \le \frac{2c_\alpha H \log K}{K},$$

where the first line invokes Lemma 1, (85) and the fact $\|\pi_{i,h}^k(s)\|_1 = 1$. Another observation is that

$$
\begin{aligned}
W_1 = \sum_{k=1}^K \big(\alpha_k^K\big)^2 \mathsf{Var}_{h,k-1}\Big(\big\langle \pi_{i,h}^k(s), \widetilde{q}_{i,h}^k(s, \cdot)\big\rangle\Big) &\le \Big\{ \max_k \alpha_k^K \Big\} \Big\{ \sum_{k=1}^K \alpha_k^K \mathsf{Var}_{h,k-1}\Big(\big\langle \pi_{i,h}^k(s), \widetilde{q}_{i,h}^k(s, \cdot)\big\rangle\Big) \Big\} \\
&\le \frac{2c_\alpha \log K}{K} \sum_{k=1}^K \alpha_k^K \mathsf{Var}_{h,k-1}\Big(\widehat{q}_{i,h}^k(s)\Big),
\end{aligned}
\tag{89}
$$

where the second line makes use of Lemma 1 and the inequality (86). With the definitions (87) and (88) in mind, invoking Freedman's inequality (i.e., Theorem 5) with $\kappa_1 = \sqrt{\frac{K \log \frac{K}{\delta}}{H}}$ then leads to

$$
\begin{aligned}
&\Big|\overline{V}_{i,h}^{\widehat{\pi}}(s) - \Big(r_{i,h}^{\widehat{\pi}}(s) + \big\langle P_{i,h}^{\widehat{\pi}}(s, \cdot), \overline{V}_{i,h+1}^{\widehat{\pi}}\big\rangle\Big)\Big| \\
&\quad = \Big| \sum_{k=1}^K \alpha_k^K \big\langle \pi_{i,h}^k(s), \widetilde{q}_{i,h}^k(s, \cdot)\big\rangle - \sum_{k=1}^K \mathbb{E}_{h,k-1}\Big[\alpha_k^K \big\langle \pi_{i,h}^k(s), \widetilde{q}_{i,h}^k(s, \cdot)\big\rangle\Big] \Big| \\
&\quad \le \kappa_1 W_1 + \Big(\frac{2}{\kappa_1} + 5R_1\Big) \log \frac{3K}{\delta}
\end{aligned}
$$

$$\leq 2c_\alpha \sqrt{\frac{\log^3 \frac{K}{\delta}}{KH}} \sum_{k=1}^{K} \alpha_k^K \mathsf{Var}_{h,k-1}\left(\widehat{q}_{i,h}^k(s)\right) + \left(2\sqrt{\frac{H}{K\log \frac{K}{\delta}}} + \frac{10c_\alpha H \log K}{K}\right) \log \frac{3K}{\delta}$$

$$\leq 2c_\alpha \sqrt{\frac{\log^3 \frac{K}{\delta}}{KH}} \sum_{k=1}^{K} \alpha_k^K \mathsf{Var}_{h,k-1}\left(\widehat{q}_{i,h}^k(s)\right) + 4\sqrt{\frac{H\log \frac{3K}{\delta}}{K}} \tag{90}$$

with probability at least $1 - \delta$, where the last relation holds true under Condition (38).

To continue, we note the first term in (90) can be bounded by Cauchy-Schwarz as follows:

$$\sum_{k=1}^{K} \alpha_k^K \mathsf{Var}_{h,k-1}\left(\widehat{q}_{i,h}^k(s)\right) = \sum_{k=1}^{K} \alpha_k^K \mathbb{E}_{h,k-1}\left[\left(\widehat{q}_{i,h}^k(s)\right)^2\right] - \sum_{k=1}^{K} \alpha_k^K \left(\mathbb{E}_{h,k-1}\left[\widehat{q}_{i,h}^k(s)\right]\right)^2$$

$$\leq \sum_{k=1}^{K} \alpha_k^K \mathbb{E}_{h,k-1}\left[\left(\widehat{q}_{i,h}^k(s)\right)^2\right] - \left(\sum_{k=1}^{K} \alpha_k^K \mathbb{E}_{h,k-1}\left[\widehat{q}_{i,h}^k(s)\right]\right)^2. \tag{91}$$

Further, we make note of two additional facts:

- The weighted mean of $\widehat{q}_{i,h}^k(s)$ obeys

$$\sum_{k=1}^{K} \alpha_k^K \mathbb{E}_{h,k-1}\left[\widehat{q}_{i,h}^k(s)\right] = \sum_{k=1}^{K} \alpha_k^K \mathbb{E}_{\boldsymbol{a}\sim\pi_h^k(s)}\left[r_{i,h}(s,\boldsymbol{a})\right] + \sum_{k=1}^{K} \alpha_k^K \mathbb{E}_{\boldsymbol{a}\sim\pi_h^k(s)}\left[\left\langle P_{i,h}(\cdot\,|\,s,\boldsymbol{a}), \overline{V}_{i,h+1}^{\widehat{\pi}}\right\rangle\right]$$

$$= r_{i,h}^{\widehat{\pi}}(s) + \left\langle P_{i,h}^{\widehat{\pi}}(s,\cdot), \overline{V}_{i,h+1}^{\widehat{\pi}}\right\rangle \geq \left\langle P_{i,h}^{\widehat{\pi}}(s,\cdot), \overline{V}_{i,h+1}^{\widehat{\pi}}\right\rangle. \tag{92}$$

- Regarding the square of $\widehat{q}_{i,h}^k(s)$, one has (see (84))

$$\left(\widehat{q}_{i,h}^k(s)\right)^2 = \left(r_{i,h}^k(s,a_{k,s}) + \left\langle P_{i,h}^k(\cdot\,|\,s,a_{k,s}), \overline{V}_{i,h+1}^{\widehat{\pi}}\right\rangle\right)^2$$

$$= \left(\left\langle P_{i,h}^k(\cdot\,|\,s,a_{k,s}), \overline{V}_{i,h+1}^{\widehat{\pi}}\right\rangle\right)^2 + \left(r_{i,h}^k(s,a_{k,s})\right)^2 + 2r_{i,h}^k(s,a_{k,s})\left\langle P_{i,h}^k(\cdot\,|\,s,a_{k,s}), \overline{V}_{i,h+1}^{\widehat{\pi}}\right\rangle$$

$$\leq \left(\left\langle P_{i,h}^k(\cdot\,|\,s,a_{k,s}), \overline{V}_{i,h+1}^{\widehat{\pi}}\right\rangle\right)^2 + 3H$$

$$\leq \left\langle P_{i,h}^k(\cdot\,|\,s,a_{k,s}), \overline{V}_{i,h+1}^{\widehat{\pi}} \circ \overline{V}_{i,h+1}^{\widehat{\pi}}\right\rangle + 3H,$$

where we have used the fact that $\|\overline{V}_{i,h+1}^{\widehat{\pi}}\|_\infty \leq H$ and $\|r_{i,h}^k\|_\infty \leq 1$; consequently,

$$\sum_{k=1}^{K} \alpha_k^K \mathbb{E}_{h,k-1}\left[\left(\widehat{q}_{i,h}^k(s)\right)^2\right] \leq \sum_{k=1}^{K} \alpha_k^K \mathbb{E}_{h,k-1}\left[\left\langle P_{i,h}^k(\cdot\,|\,s,a_{k,s}), \overline{V}_{i,h+1}^{\widehat{\pi}} \circ \overline{V}_{i,h+1}^{\widehat{\pi}}\right\rangle\right] + 3H$$

$$= \sum_{k=1}^{K} \alpha_k^K \sum_{a_i\in\mathcal{A}_i} \pi_{i,h}^k(a_i\,|\,s)\mathbb{E}_{h,k-1}\left[\left\langle P_{i,h}^k(\cdot\,|\,s,a_i), \overline{V}_{i,h+1}^{\widehat{\pi}} \circ \overline{V}_{i,h+1}^{\widehat{\pi}}\right\rangle\right] + 3H$$

$$= \left\langle P_{i,h}^{\widehat{\pi}}(s,\cdot), \overline{V}_{i,h+1}^{\widehat{\pi}} \circ \overline{V}_{i,h+1}^{\widehat{\pi}}\right\rangle + 3H. \tag{93}$$

Taking (92) and (93) together with (91) yields

$$\sum_{k=1}^{K} \alpha_k^K \mathsf{Var}_{h,k-1}\left(\widehat{q}_{i,h}^k(s)\right) \leq \sum_{k=1}^{K} \alpha_k^K \mathbb{E}_{h,k-1}\left[\left(\widehat{q}_{i,h}^k(s)\right)^2\right] - \left(\sum_{k=1}^{K} \alpha_k^K \mathbb{E}_{h,k-1}\left[\widehat{q}_{i,h}^k(s)\right]\right)^2$$

$$\leq \left\langle P_{i,h}^{\widehat{\pi}}(s,\cdot), \overline{V}_{i,h+1}^{\widehat{\pi}} \circ \overline{V}_{i,h+1}^{\widehat{\pi}}\right\rangle - \left(\left\langle P_{i,h}^{\widehat{\pi}}(s,\cdot), \overline{V}_{i,h+1}^{\widehat{\pi}}\right\rangle\right)^2 + 3H.$$

To finish up, substituting these into (90) and making use of the assumption (38) give

$$\left|\overline{V}_{i,h}^{\widehat{\pi}}(s) - \left(r_{i,h}^{\widehat{\pi}}(s) + \left\langle P_{i,h}^{\widehat{\pi}}(s,\cdot), \overline{V}_{i,h+1}^{\widehat{\pi}}\right\rangle\right)\right|$$

$$\leq 2c_\alpha \sqrt{\frac{\log^3 \frac{K}{\delta}}{KH}} \left[ \left\langle P_{i,h}^{\widehat{\pi}}(s,\cdot), \overline{V}_{i,h+1}^{\widehat{\pi}} \circ \overline{V}_{i,h+1}^{\widehat{\pi}} \right\rangle - \left( \left\langle P_{i,h}^{\widehat{\pi}}(s,\cdot), \overline{V}_{i,h+1}^{\widehat{\pi}} \right\rangle \right)^2 \right] + (6c_\alpha + 4) \sqrt{\frac{H \log^3 \frac{K}{\delta}}{K}}$$

for any $s \in \mathcal{S}$, thus concluding the proof of the first claim (49a) of Lemma 4.

The second claim (49b) of Lemma 4 can be established using exactly the same argument, and hence we omit the proof here for the sake of brevity.

### E.2.2  Proof of inequality (49c)

We then turn to the last advertised inequality (49c). Given that $\overline{r}_{i,h}(s) + \overline{P}_{i,h}(s,\cdot)\widehat{V}_{i,h+1} \in [0, H - h + 1]$ for all $s \in \mathcal{S}$, we can recall the definition (20a) of $\widehat{V}_{i,h}$ to obtain

$$\left| \widehat{V}_{i,h}(s) - \left( \overline{r}_{i,h}(s) + \overline{P}_{i,h}(s,\cdot)\widehat{V}_{i,h+1} \right) \right| \leq \left| \sum_{k=1}^{K} \alpha_k^K \left\langle \pi_{i,h}^k(\cdot \mid s), q_{i,h}^k(s,\cdot) \right\rangle + \beta_{i,h}(s) - \left( \overline{r}_{i,h}(s) + \overline{P}_{i,h}(s,\cdot)\widehat{V}_{i,h+1} \right) \right|$$
(94)

for all $s \in \mathcal{S}$. The remaining analysis is dedicated to bounding the right-hand side of (94).

Let us begin with the following identity:

$$\sum_{k=1}^{K} \alpha_k^K \left\langle \pi_{i,h}^k(\cdot \mid s), q_{i,h}^k(s,\cdot) \right\rangle + \beta_{i,h}(s) = \sum_{k=1}^{K} \alpha_k^K \mathop{\mathbb{E}}_{a_i \sim \pi_{i,h}^k(s)} \left[ r_{i,h}^k(s,a_i) + P_{i,h}^k(\cdot \mid s, a_i)\widehat{V}_{i,h+1} \right] + \beta_{i,h}(s)$$
$$= \overline{r}_{i,h}(s) + \left\langle \overline{P}_{i,h}(s,\cdot), \widehat{V}_{i,h+1} \right\rangle + \beta_{i,h}(s),$$
(95)

where we recall the definitions of $\overline{r}_{i,h} \in \mathbb{R}^S$ and $\overline{P}_{i,h} \in \mathbb{R}^{S \times S}$ in (48). The key step boils down to bounding the bonus term defined in (24), towards which we first claim that

$$\sum_{k=1}^{K} \alpha_k^K \mathsf{Var}_{\pi_{i,h}^k(s)} \left( q_{i,h}^k(s,\cdot) \right) \leq 2 + 2 \left[ \overline{P}_{i,h}(s,\cdot)\left( \widehat{V}_{i,h+1} \circ \widehat{V}_{i,h+1} \right) - \left( \overline{P}_{i,h}(s,\cdot)\widehat{V}_{i,h+1} \right)^2 \right] \quad (96)$$

holds for all $s \in \mathcal{S}$. Assuming the validity of this claim, we can then demonstrate that

$$\beta_{i,h}(s) = c_{\mathsf{b}} \sqrt{\frac{\log^3 \left( \frac{KS \sum_i A_i}{\delta} \right)}{KH}} \sum_{k=1}^{K} \alpha_k^K \left\{ \mathsf{Var}_{\pi_{i,h}^k(s)} \left( q_{i,h}^k(s,\cdot) \right) + H \right\}$$

$$\leq 2c_{\mathsf{b}} \sqrt{\frac{\log^3 \left( \frac{KS \sum_i A_i}{\delta} \right)}{KH}} \left\{ \overline{P}_{i,h}(s,\cdot)\left( \widehat{V}_{i,h+1} \circ \widehat{V}_{i,h+1} \right) - \left( \overline{P}_{i,h}(s,\cdot)\widehat{V}_{i,h+1} \right)^2 + H \right\},$$
(97)

where we have used the identity $\sum_{k=1}^{K} \alpha_k^K = 1$. Hence, we can readily establish the desired result (49c) by combining (97) with (95) and (94), provided that $c_3 > 0$ is sufficiently large.

It remains to justify the claim (96). Towards this end, we make the observation that

$$\mathsf{Var}_{\pi_{i,h}^k(s)} \left( q_{i,h}^k(s,\cdot) \right) \leq 2\mathsf{Var}_{\pi_{i,h}^k(s)} \left( r_{i,h}^k(s,\cdot) \right) + 2\mathsf{Var}_{\pi_{i,h}^k(s)} \left( \sum_{s'} P_{i,h}^k(s' \mid s, \cdot)\widehat{V}_{i,h+1}(s') \right)$$

$$\leq 2 + 2 \left[ \sum_{a_i} \pi_{i,h}^k(a_i \mid s) P_{i,h}^k(\cdot \mid s, a_i)\left( \widehat{V}_{i,h+1} \circ \widehat{V}_{i,h+1} \right) - \left( \sum_{a_i} \pi_{i,h}^k(a_i \mid s) P_{i,h}^k(\cdot \mid s, a_i)\widehat{V}_{i,h+1} \right)^2 \right],$$

which results from $\|r_{i,h}^k\|_\infty \leq 1$ and the following relation:

$$\mathsf{Var}_{\pi_{i,h}^k(s)} \left( \sum_{s'} P_{i,h}^k(s' \mid s, \cdot)\widehat{V}_{i,h+1}(s') \right)$$

$$= \sum_{a_i} \pi_{i,h}^k(a_i \mid s) \left( P_{i,h}^k(\cdot \mid s, a_i)\widehat{V}_{i,h+1} \right)^2 - \left( \sum_{a_i} \pi_{i,h}^k(a_i \mid s) P_{i,h}^k(\cdot \mid s, a_i)\widehat{V}_{i,h+1} \right)^2$$

$$\leq \sum_{a_i} \pi_{i,h}^k(a_i \mid s) P_{i,h}^k(\cdot \mid s, a_i)\left( \widehat{V}_{i,h+1} \circ \widehat{V}_{i,h+1} \right) - \left( \sum_{a_i} \pi_{i,h}^k(a_i \mid s) P_{i,h}^k(\cdot \mid s, a_i)\widehat{V}_{i,h+1} \right)^2.$$

This taken together with the fact $\sum_{k=1}^K \alpha_k^K = 1$ and Jensen's inequality yields

$$\sum_{k=1}^K \alpha_k^K \mathsf{Var}_{\pi_{i,h}^k(s)}\big(q_{i,h}^k(s,\cdot)\big)$$

$$\leq \sum_{k=1}^K \alpha_k^K \left\{ 2 + 2\left[ \sum_{a_i} \pi_{i,h}^k(a_i \mid s) P_{i,h}^k(\cdot \mid s, a_i)\big(\widehat{V}_{i,h+1} \circ \widehat{V}_{i,h+1}\big) - \bigg( \sum_{a_i} \pi_{i,h}^k(a_i \mid s) P_{i,h}^k(\cdot \mid s, a_i)\widehat{V}_{i,h+1} \bigg)^2 \right] \right\}$$

$$\leq 2 + 2\overline{P}_{i,h}(s,\cdot)\big(\widehat{V}_{i,h+1} \circ \widehat{V}_{i,h+1}\big) - 2\bigg( \sum_{k=1}^K \alpha_k^K \sum_{a_i} \pi_{i,h}^k(a_i \mid s) P_{i,h}^k(\cdot \mid s, a_i)\widehat{V}_{i,h+1} \bigg)^2$$

$$= 2 + 2\Big[ \overline{P}_{i,h}(s,\cdot)\big(\widehat{V}_{i,h+1} \circ \widehat{V}_{i,h+1}\big) - \big(\overline{P}_{i,h}(s,\cdot)\widehat{V}_{i,h+1}\big)^2 \Big]$$

as claimed.

### E.3 Minimax lower bound

In this section, we formalize the minimax lower bound claimed in (31).

**Theorem 4** (Minimax lower bound). *Consider any $m \geq 2$ and any $0 < \varepsilon \leq c_1 H$ for some small enough constant $c_1 > 0$. Then one can construct a collection of $m$-player zero-sum Markov games $\{\mathcal{MG}_\theta \mid \theta \in \Theta\}$ with $S$ states, horizon $H$, and $A_i$ actions for the $i$-th player $(1 \leq i \leq m)$ such that*

$$\inf_{\widehat{\pi}} \max_{\theta \in \Theta} \mathbb{P}^{\mathcal{MG}_\theta}\big\{ \mathsf{gap}(\widehat{\pi}) > \varepsilon \big\} \geq \frac{1}{4}, \tag{98}$$

*provided that the total sample size obeys*

$$N \leq \frac{c_2 H^4 S \max_{1 \leq i \leq m} A_i}{\varepsilon^2} \tag{99}$$

*for some sufficiently small constant $c_2 > 0$. Here, the infimum is over all (joint) policy estimator $\widehat{\pi}$, and $\mathbb{P}^{\mathcal{MG}_\theta}$ denotes the probability when the Markov game is $\mathcal{MG}_\theta$.*

*Proof.* Suppose without loss of generality that $A_1 \geq \max\{A_2, \ldots, A_m\}$. Let us begin by considering the special scenario with $A_2 = \ldots = A_m = 1$; in this case, computing either the NE or the CCE reduces to finding the optimal policy of a single-agent MDP with $S$ states and $A_1$ actions. It is well-known that for any given accuracy level $\varepsilon \in (0, H]$, there exists a non-stationary MDP with $S$ states and $A_1$ actions such that no algorithm can learn an $\varepsilon$-optimal policy with $o\big(\frac{H^4 S A_1}{\varepsilon^2}\big)$ samples [3, 41]. More precisely, for any given $0 < \varepsilon \leq c_1 H$ for some small enough constant $c_1 > 0$, one can construct a collection of MDPs $\{\mathcal{M}_\theta \mid \theta \in \Theta\}$ such that

$$\inf_{\widehat{\mu}} \max_{\theta \in \Theta} \mathbb{P}^{\mathcal{M}_\theta} \left\{ \max_{s \in \mathcal{S}} \big(V_1^\star(s) - V_1^{\widehat{\mu}}(s)\big) > \varepsilon \right\} \geq \frac{1}{4}, \tag{100}$$

with the proviso that the total sample size obeys

$$N \leq \frac{c_2 H^4 S A_1}{\varepsilon^2} \tag{101}$$

for some small enough constant $c_2 > 0$. Here, the infimum is over all policy estimate $\widehat{\mu}$ in this single-agent scenario, and $\mathbb{P}^{\mathcal{M}_\theta}$ denotes the probability when the MDP is $\mathcal{M}_\theta$.

Next, let us construct a collection of Markov games by augmenting each of the single-agent MDPs $\mathcal{M}_\theta$ with $A_i$ completely identical actions for the $i$-th player $(2 \leq i \leq m)$; that is, to construct $\mathcal{MG}_\theta$, we take its reward function and probability transition kernel to be

$$r_{i,h}^{\mathcal{MG}_\theta}(s, \boldsymbol{a}) = \begin{cases} r_h^{\mathcal{M}_\theta}(s, a_1) & \text{if } i = 1 \\ -r_h^{\mathcal{M}_\theta}(s, a_1) & \text{if } i = m \\ 0 & \text{else} \end{cases} \quad \text{and} \quad P_h^{\mathcal{MG}_\theta}(\cdot \mid s, \boldsymbol{a}) = P_h^{\mathcal{M}_\theta}(\cdot \mid s, a_1) \tag{102}$$

for all $(s, h, \boldsymbol{a} = [a_1, \ldots, a_m]) \in \mathcal{S} \times [H] \times \mathcal{A}$. Evidently, finding either an NE or a CCE of $\mathcal{MG}_\theta$ is equivalent to computing the optimal policy of $\mathcal{M}_\theta$, given the non-distinguishability of the actions of all but the first player in $\mathcal{MG}_\theta$. This in turn immediately establishes the advertised lower bound. $\square$

### E.4 Freedman's inequality

In this section, we record the Freedman inequality for martingales [27] with slight modification, which is a crucial concentration bound for our analysis.

**Theorem 5.** *Suppose that $Y_n = \sum_{k=1}^{n} X_k \in \mathbb{R}$, where $\{X_k\}$ is a real-valued scalar sequence obeying*

$$|X_k| \leq R \qquad and \qquad \mathbb{E}\left[X_k \mid \{X_j\}_{j:j<k}\right] = 0 \qquad for\ all\ k \geq 1$$

*for some quantity $R > 0$. Define*

$$W_n := \sum_{k=1}^{n} \mathbb{E}_{k-1}\left[X_k^2\right],$$

*where $\mathbb{E}_{k-1}$ stands for the expectation conditional on $\{X_j\}_{j:j<k}$. Consider any arbitrary quantity $\kappa > 0$. With probability at least $1 - \delta$, one has*

$$|Y_n| \leq \sqrt{8W_n \log \frac{3n}{\delta}} + 5R \log \frac{3n}{\delta} \leq \kappa W_n + \left(\frac{2}{\kappa} + 5R\right) \log \frac{3n}{\delta}. \tag{103}$$

*Proof.* Suppose that $W_n \leq \sigma^2$ holds deterministically for some quantity $\sigma^2$. As has been demonstrated in Li et al. [39, Theorem 5], with probability at least $1 - \delta$ we have

$$|Y_n| \leq \sqrt{8 \max\left\{W_n, \frac{\sigma^2}{2^K}\right\} \log \frac{2K}{\delta}} + \frac{4}{3} R \log \frac{2K}{\delta} \tag{104}$$

for any positive integer $K \geq 1$. Recognizing the trivial bound $W_n \leq nR^2$, one can take $\sigma^2 = nR^2$ and $K = \log_2 n$ to obtain

$$
\begin{aligned}
|Y_n| &\leq \sqrt{8 \max\left\{W_n, R^2\right\} \log \frac{4\log_2 n}{\delta}} + \frac{4}{3} R \log \frac{4\log_2 n}{\delta} \\
&\leq \sqrt{8W_n \log \frac{3n}{\delta}} + \sqrt{8R^2 \log \frac{3n}{\delta}} + \frac{4}{3} R \log \frac{3n}{\delta} \\
&\leq \sqrt{8W_n \log \frac{3n}{\delta}} + 5R \log \frac{3n}{\delta},
\end{aligned}
$$

where we have used $4\log_2 n \leq 3n$ for any integer $n \geq 1$. This establishes the first inequality in (103). The second inequality in (103) is then a direct consequence of the elementary inequality $2ab \leq a^2 + b^2$. $\qquad\square$