# OpenReview forum: "Minimax-Optimal Multi-Agent RL in Markov Games With a Generative Model"
_NeurIPS.cc/2022/Conference — NeurIPS 2022 Accept_

### Official Review · Reviewer_VUwp · 2022-07-06

**Rating:** 7
**Confidence:** 3
**Soundness:** 3 good
**Presentation:** 3 good
**Contribution:** 3 good

**Summary:**

This paper provides a new algorithm for learning a Nash equilibrium in a zero-sum Markov game and proves that the sample complexity is much tighter than in the previous works, in a sense that the bound is nearly optimal.

**Questions:**

None.

**Limitations:**

The limitations have been stated clearly in the paper.

**Strengths And Weaknesses:**

Strengths:
1. The algorithm and the theoretical results seem novel and appealing. As stated in the paper, it is proven that the proposed algorithm overcomes (i) the curse of multiple agents, and (ii) the curse of horizon. The results seem sound and significant.
2. The paper is well structured and well written. The notation is clear and easy to follow.

Weaknesses:
1.  The "main results" section is short, and the theoretical results are not explained clearly. How do we get to Theorem 1? Since this is a theoretical paper, I think an outline of the proof is needed here to help the readers to get the idea quickly.
2. Eq. (27) makes sense to me. However, it is a bit risky to say that the bound proven in the paper is "nearly improvable" without proof.

---

> ### Author Response · Authors · 2022-08-02
> **Thanks for the positive evaluation of our paper as well as the valuable questions.  Below is a point-to-point response to the reviewer's questions.**
>
> 1. Indeed, the "main results" section is somewhat short without including a proof sketch, mainly due to page limitations.
> To address the reviewer's concerns,
> >- we have included a high-level description of the main proof ideas in the supplementary material; see the beginning of Appendix C.2 in the revised version.
> Following such high-level ideas, we further include an expanded step-by-step descriptions of these steps also in Appendix C.2 in the revised version, with more technical details postponed to later sections.
> If the paper is accepted and if one more page is allowed in the final version (as in last year), we will include the high-level proof sketch in the main text as well.
> >- we have also further simplified the algorithm (by eliminating the complicated upper confidence bonus originally employed in the Q-function estimation); see Section 3.1 of the revised paper. Hopefully this also helps improve a bit the readability of the algorithm.
>
> 2. Thanks for pointing out the lack of details about the minimax lower bound claimed in Eq. (27). The proof of this result is based on reduction to the minimax lower bound for single-agent MDPs [1, 2]. To address the reviewer's concerns, we have included a formal theorem statement of this lower bound along with its proof in Appendix G in the supplementary material of the revised paper.
>
> 2. Thanks for pointing out the lack of details about the minimax lower bound claimed in Eq. (27). The proof of this result is based on reduction to the minimax lower bound for single-agent MDPs [1, 2]. To address the reviewer's concerns, we have included a formal theorem statement of this lower bound along with its proof in Appendix G in the supplementary material of the revised paper.
>
>
> References:
>
> [1] Minimax PAC bounds on the sample complexity of reinforcement learning with a generative model,
> M. G. Azar, R. Munos, and H. Kappen, 2013.
>
> [2] Settling the sample complexity of model-based offline reinforcement learning,
> G. Li, L. Shi, Y. Chen, Y. Chi, and Y. Wei, 2022.

---

### Official Review · Reviewer_1o3R · 2022-07-11

**Rating:** 7
**Confidence:** 4
**Soundness:** 3 good
**Presentation:** 3 good
**Contribution:** 4 excellent

**Summary:**

This work focuses on closing the sample complexity gap in learning two-player zero-sum Markov games. With previous results suffering either the curse of multi-agent or a loose horizon-dependency, this work breaks both barriers simultaneously in the setting of a generative model and approaches the optimal rate up to logarithmic factors. The algorithm design benefits from two core ideas: the optimism to tackle the uncertainty of the environment and the adversarial learning (FTRL in particular) to overcome the non-stationary environment from each player's perspective. The core technique that contributes to the tight sample complexity is the data-driven Bernstein-style bonus.

**Questions:**

**I am overall very satisfied with this work. The following questions are more from my own curiosities.**
- [27] mentions that its result is loose due to two reasons (see the illustration below their Thm 4). One reason is the incremental update and use of the learning rate therein. Another reason is that [27] did not have a variance-aware analysis. The improvement over the second point is clear in this work. I am wondering how this work addresses the first point. Especially, is the generative model alleviate the first point directly? Or the learning rate chosen in this work (different from which in [27]) plays a critical role?
- Regarding the variance-aware design, I noticed that it is considered in both the confidence-bound design and the learning rates for FTRL (i.e., $\eta$). I am wondering about the reason for each individual consideration. Hopefully, the authors can give some intuitive explanations.
- Additionally, for the confidence bound, the update of the Q value and the V value uses two different confidence bounds. They share similar formats but it will still be helpful if the authors can illustrate their relationships.
- Another question regarding this work and [27] is that the output policy in [27] is typically non-Markovian, I am wondering whether the authors can give some illustration on why this work can output a Markov policy. I did not find much emphasis on this point and wonder whether there is any particular reason that outputting a Markov policy is straightforward in this work.

**Limitations:**

Yes, the authors have addressed the limitation of this work and there is no major negative societal impact given its theoretical nature.

**Strengths And Weaknesses:**

- (S) While many advances in learning in MGs have been made in recent years, how to achieve the optimal sample complexity remains open even in the "simplest" case of two-player zero-sum games and generative models, as considered in this work. The progress of this work largely contributes to a deeper understanding of this problem and should inspire future works.
- (S) The techniques utilized in this work nicely improve previous ones. While optimism and adversarial learning in MG have been investigated in MGs [27], this work nicely spots that the looseness of the previous result is due to the failure of considering variance-type bounds. While similar ideas of using Bernstein-type bounds to sharpen the horizon-dependency are philosophically well-recognized in RL, this work still skilfully implements it in the context of MGs and combines it with adversarial learning.
- (S) While this work is heavy in theory, the major design intuitions are given and the results are illustrated clearly, which is a joy to read.
- (S) The tools developed in this work, especially the variance-type bound for FTRL should be of independent usage and may benefit other studies beyond MGs.
- (W) If have to say something about weakness, the design idea of this work is similar to that of [27] although this work successfully incorporated the variance-type analysis (which is lacking in [27]). Also, the assumption of the generative model is somewhat limited. However, I still believe that compared with [27], sufficient contributions are made even assuming the generative model.

---

> ### Author Response · Authors · 2022-08-02
> **We thank the reviewer for careful reading of the paper and for insightful and valuable feedbacks, which have led to multiple simplifications of our algorithm design. Please find below our point-to-point response.**
>
> 1. As the reviewer pointed out, our improvement is due to two key factors:
> >- our algorithm adopts the variance-aware Bernstein-style bonus design, which typically improve upon the Hoeffding-type bonus by a factor of $H$;
> >- the use of model-free type algorithms in [27] --- in conjunction with the learning rates used therein --- also contributes to another loss of $O(H)$ factor, similar to existing analysis for Q-learning in the absence of variance reduction.
> >
> In fact, with our carefully chosen learning rates and the fact that we can settle each layer $h$ completely before moving backward,
> our Q-estimation process for each layer $h$
> shares some similar spirit with a model-based algorithm, as it is able to make effective use of most samples to implicitly estimate models;  this stands in stark contrast to the model-free type paradigm in [27] that effectively focuses only on the more recent portion of samples due to the use of more aggressive learning rates.  We will discuss this in more detail in the revised paper.
>
>
> 2. Thanks much for raising the question regarding the role of variance-aware design in the learning rates for FTRL.
> Much inspired by the reviewer's question, we rethink our algorithm design and analysis and realize that
> the learning rates for FTRL can be chosen in a much simpler way (i.e., $\eta_{k+1}=\sqrt{\frac{\log K}{\alpha_k H}}$)
> without the need of incorporating variance-style quantities.
> See Equation (20) and the revised proof in the revision.
> This leads to a simpler paradigm that is easier to understand than in the original submission.  In the meantime, the variance-aware choice in the upper confidence bound during value estimation
> --- which needs to be carefully designed in order to allow for certain variance decomposability across time steps ---
> adopts the Bernstein-type design principle.
>
> 3. We are also grateful to the reviewer for the question regarding the adoption of upper confidence bounds in Q-function estimates,
> which inspires another remarkable simplification of our algorithm design.
> More specifically, after carefully rethinking the necessity of this part,
> we are now able to completely get rid of the UCB-type terms when updating Q-functions,
> using a more delicate analysis to take care of the uncertainty incurred in this stage.
> See Equations (13)-(15) for a cleaner algorithm design as well as the new proof in the revision.
> In summary, now we only need to incorporate UCB-type bonuses when generating the final value function estimates for each step $h$.
> On the other hand, the incorporation of UCB in final value estimation seems crucial in the current analysis,
> which is employed to compensate for the uncertainty resulting from the difference of variances across actions as well as the time-varying policy iterates of the opponent.
>
>
> 4. Our algorithm is able to output a Markovian policy (depending only on the current $s$ and $h$) largely thanks to the availability of the generative model.
> Briefly speaking, the presence of the generative model allows us to work completely backward (in terms of both the sampling process and the algorithm); namely, we can settle the policy learning process for the $(h+1)$-th layer completely  before moving backward to cope with the $h$-th layer. In contrast, this is typically not feasible in the online exploration case, as sampling is done in an episode-based manner and the policy estimates for later layers depend heavily on the previous layers.
>
>
> 5. Finally, we would like to note that our current analysis framework can be extended beyond generative models (although it is not yet ready to accommodate the fully online exploration setting).  For instance, similar ideas are effective for the so-called local access model [1] or the state-revisiting model [2],
> where we are allowed to revisit a previously visited state and generate a new (partial) episode from there (without completely restarting from a new state in the first layer). Another highlight and challenging aspect of our result is the development of variance-type analysis for FTRL,
> which is crucial in enabling optimal variance-aware design in solving Markov games.
>
>
> References:
>
> [1] Efficient local planning with linear function approximation,
> Yin, D., Hao, B., Abbasi-Yadkori, Y., Lazic, N., and Szepesvari, C, 2021.
>
> [2] Sample-Efficient Reinforcement Learning Is Feasible for Linearly Realizable MDPs with Limited Revisiting,
> G. Li, Y. Chen, Y. Chi, Y. Gu, and Y. Wei, 2021.

---

> > ### Comment · Reviewer_1o3R · 2022-08-08
> > **Thank you for the responses!**
> >
> > I sincerely thank the authors for the responses and much appreciate the modifications during the revisions. It is a great pleasure to participate in the discussions and see that my comments contribute to this work. I would keep and defend my positive opinion of this work.

---

### Official Review · Reviewer_H3Nw · 2022-07-11

**Rating:** 7
**Confidence:** 3
**Soundness:** 3 good
**Presentation:** 3 good
**Contribution:** 3 good

**Summary:**

This paper studies the problem of finding the Nash equilibria in two-player zero-sum Markov games with a generative model. The authors established a minimax optimal sample complexity in this setting.


**Questions:**

1. The previous method (V-learning) only updates V functions and thus breaks the curse of multiagents. So I am curious about that why the proposed algorithm, which updates Q functions, can break the curse of multiagents.

2. Because the proof is too long, I did not read the proof in its entirety. Can the authors briefly describe the general idea of ​​the proof? I also encourage the authors to add a proof sketch.

3. I hope the authors make more comments on why this method cannot be applied to the online setting.

I will consider increasing the score if the authors can address the above concerns.

**Limitations:**

Yes

**Strengths And Weaknesses:**

 I think this problem is very interesting for the RL committee. But perhaps for too technical reasons, the paper is hard to follow. I hope the authors can make more comments to make the paper clearer. See the questions below for details.

---

> ### Author Response · Authors · 2022-08-02
> **Thanks a lot for the helpful comments and valuable feedback. Please find below our point-to-point response, which will also be incorporated into the final paper.**
>
> 1. Let us provide further intuitions/explanations regarding how we manage to break the curse of multiagents,
> which we will detail in the final paper. An important aspect worth highlighting is that:
> although we conduct Q-function estimation, the Q-function considered here is concerned with the one of only {\em a single player},
> which plays a critical role in overcoming the curse of multiple agents.  For instance, we are effectively estimating the Q-function $Q_h^{\nu}(s_h, a_h)$ of the max-player under the near-optimal policy of the min-player,  rather than estimating the full version of Q-function $Q_h(s_h, a_h, b_h)$ that cover all combinations of joint actions.  In comparison, the V-learning algorithm also maintains certain intermediate estimates that range over the state-action pairs for each player;
> more specifically,  for each agent and each state $s_h$ in the $h$-th step, ADV\_BANDIT\_UPDATE procedure in V-learning (see Protocol 2) needs to estimate the accumulated loss function $\ell(a_h)$ for all actions (each of this actually corresponds to $1 - \frac{1}{H}Q_h(s_h, a_h)$). In summary, the key to breaking the curse of multiagents is to only maintain single-agent type quantities (e.g., $f_h(s_h,a_h)$)
> rather than using multi-agent type functions (e.g., $f_h(s_h,a_h,b_h)$).
> The revised paper has also clarified the importance of this point right after the definition of these estimates, which reads:
> *"Crucially, the above objects are all constructed from the perspective of a single player (either the max-player or the min-player), and are all quantities needed to operate a "single-agent" MDP (as opposed to MARL).  As such, the complexity of storing/updating the above objects only scales with the aggregate size of the individual action space, rather than the size of the product action space. This is a key point that allows one to overcome the curse of multiple agents."*
>
>
> 2. The main text of our original submission did not include a proof sketch mainly due to page limitations.
> To address the reviewer's question,
>  we have included (i) a short high-level summary of the proof at the beginning of Appendix C.2 in the revision;
>  (ii) an expanded step-by-step description of the main steps of the proof in Appendix C.2 of the revised version.
> These taken together present the key ideas underlying the proof.
>  If one more page is allowed in the final version (as in last year),
>  then we will move the short version of the proof outline to the main text of the paper.
>
>
>
> 3. Thanks for the question regarding the applicability of our results to the online RL setting.
> We have two points to make regarding this, which we shall elaborate on in the final version as well.
> >- On one hand, it is indeed possible to invoke some of our algorithmic ideas to help improve the performance guarantees for the online setting. For instance, the current result for V-learning is suboptimal in terms of $H$ for at least two reasons:
> 		(i) V-learning employs the Hoeffding-type bonus, which is known to be suboptimal even for the single-agent case;
> 		(ii) Given the model-free nature of V-learning, the learning rates adopted therein are overly aggressive,
> 		which focus too much on the more recent portion of data without making the most efficient use of early data (in fact, this issue has been recognized in Q-learning in the single-agent setting as well).
> 		As a result, our variance-aware and data-driven design of upper confidence bounds
> 		are promising in improving the horizon dependency for the online setting (e.g., improving the state-of-the-art result by a factor of $H$).
> >- On the other hand, we are not sure whether our current analysis framework suffices to help achieve sample optimality in the online setting. As a key ingredient of our algorithm design and analysis,
> 		we always settle the estimation process for step $h+1$ completely before moving backward to perform estimation for step $h$.
> 		This is enabled largely due to the availability of the generative model.
> 		When it comes to the online setting, however, we can no longer implement this paradigm in the most data-hungry regime.
> 		Each sample episode exhibits temporal correlation, and the samples drawn for later steps are always heavily dependent on earlier samples in this episode. As a consequence, more delicate algorithmic ideas beyond the current paper are needed in order to achieve sample optimality in the online RL setting.
>
>
>
> References:
>
> [1] V-Learning--A Simple, Efficient, Decentralized Algorithm for Multiagent RL,
> C. Jin, Q. Liu, Y. Wang, and T. Yu, 2021

---

> > ### Comment · Reviewer_H3Nw · 2022-08-07
> > **Response**
> >
> > Thanks for your detailed response, which has addressed most of my concerns. So I have raised my score to 7.

---

### Meta-Review · Area_Chair_FCXx · 2022-08-26

**Recommendation:** Accept
**Confidence:** Certain

**Metareview:**

This paper received uniformly positive reviews on a topic of relevance to the theory-ML community where understanding how to achieve tight sample complexity results in Markov Games has been of significant interest in recent years.

**Award:**

No

---

### Decision · Program_Chairs · 2022-09-14

Accept